# Purcell enhancement of directional edge photocurrent in a van der Waals self-cavity

Xinyu Li [1,2], Jesse Hagelstein[1,2], Gunda Kipp [1,2], Felix Sturm [1,2,3], Kateryna Kusyak [1,2,3], Yunfei Huang[3], Benedikt Schulte [1,2,3], Alexander M. Potts [1,2,3], Jonathan Stensberg [3,4], Victoria Quirós-Cordero[5], Chiara Trovatello [5,6], Zhi Hao Peng[4], Chaowei Hu [7], Jonathan M. DeStefano [7], Michael Fechner [1,2], Takashi Taniguchi [8], Kenji Watanabe [9], P. James Schuck [5], Xiaodong Xu [7], Jiun-Haw Chu [7], Xiaoyang Zhu [4], Angel Rubio [1,2,10], Marios H. Michael[1,2], Matthew W. Day[1,2,3], Hope M. Bretscher [1,2,3] ✉ & James W. McIver [1,2,3] ✉

Cavities provide a means to manipulate the optical and electronic responses of quantum materials by selectively enhancing light-matter interaction at specific frequencies and momenta. While cavities typically involve external structures, exfoliated flakes of van der Waals (vdW) materials can form intrinsic self-cavities due to their small finite dimensions, confining electromagnetic fields into plasmonic cavity modes, characterized by standing-wave current distributions. While cavity-enhanced phenomena are well-studied at optical frequencies, the impact of self-cavities on nonlinear electronic responses—such as directional photocurrent—remains largely unexplored, particularly in the terahertz regime, critical for emerging ultrafast optoelectronic technologies. Here, we report a self-cavity-induced Purcell enhancement of directional photocurrents in the vdW semimetal WTe$_2$. Using ultrafast optoelectronic circuitry, we measured coherent near-field THz emission resulting from nonlinear photocurrents excited at the sample edges. We observed enhanced emission at finite frequencies, tunable via excitation fluence and sample geometry, which we attribute to plasmonic interference effects controlled by the cavity boundaries. We developed an analytical theory that captures the cavity resonance conditions and spectral response across multiple devices. Our findings establish WTe$_2$ as a bias-free, geometry-tunable THz emitter and demonstrate the potential of self-cavity engineering for controlling nonlinear, nonequilibrium dynamics in quantum materials.

Tailoring the electromagnetic environment of quantum materials offers a strategy for tuning macroscopic material properties and delivering new functionality[1–4]. In particular, cavities–which modify the photonic density of states–can enhance light-matter coupling, enabling phenomena such as exciton-polariton condensation[5], low-intensity Floquet engineering[6], and ultrafast optical switching[7]. While these effects have been explored at visible and near-infrared frequencies, their impact at low energies, especially in the terahertz (THz) regime relevant to collective excitations and nonlinear transport, remains largely uncharted.

Two-dimensional layered van der Waals (vdW) materials are ideal platforms to bridge this gap, combining giant nonlinearities with inherent cavity functionality. Second-order nonlinear responses have been observed to be significant in vdW systems with broken inversion symmetry[8]. For example, in semimetals like WTe$_2$, photoexcitation generates ultrafast photogalvanic currents via quantum geometric effects[9,10] and anisotropic carrier scattering[11]. These nonlinear currents are not only probes for topological electronic and structural properties, but also could be used for compact, next-generation THz technologies, including 6G communications[12]. Notably, by sustaining standing wave resonances due to interference between edge-reflected currents, micron-scale vdW flakes act as plasmonic self-cavities in the THz range without the need for external mirrors[13,14].

This dual role—as both nonlinear medium and cavity—raises a key question: how does the Purcell effect[15], a hallmark of cavity quantum electrodynamics, manifest in driven, nonlinear currents? Resolving how a cavity-engineered photonic density of states modifies ultrafast transport in a quantum material could advance fundamental understanding of finite-size effects in THz emission[12,16] and identify future routes for scalable, bias-free THz sources for sensing and communication.

In this work, we demonstrate that plasmonic self-cavities in WTe$_2$ can enhance THz photocurrent in photoexcited WTe$_2$ self-cavities. Using THz optoelectronic circuitry, we detected near-field THz emission consistent with directional photocurrents, as illustrated in Fig. 1a. A resonance appears when the device is photoexcited outside the co-planar stripline, which we attribute to a Purcell enhancement at the self-cavity resonance frequency due to the cavity-engineered photonic density (see Fig. 1b). We developed and applied an analytical theory to describe tunable self-cavity modes that reproduce experimental trends. These results show that cavity engineering can be applied to driven nonlinear currents, establishing a platform for bias-free, geometry-tunable THz emission from quantum materials.

## Results

### Device fabrication and signal readout

VdW heterostructures of hBN-encapsulated few-layer T$_d$-WTe$_2$ were interfaced with THz optoelectronic circuitry[13,17–20]. Each heterostructure was fabricated by mechanically exfoliating flakes of T$_d$-WTe$_2$, transferring them to a sapphire substrate, and encapsulating them in hBN. We measured four devices (A-D) with varying flake thicknesses

and orientations, as shown in Fig. 2 and Supplementary Fig. 3. A co-planar stripline was patterned on top of each device. Photoexcitation of the heterostructure with linearly polarized, 515 nm pulses of 100 fs light at normal incidence generated a transient current in the device. Radiation from current in the region between the stripline traces (denoted as region 2, see Supplementary Fig. 13) is coupled into the odd-mode of the co-planar stripline, with the electric field pointing from one gold trace to the other[12,21,22]. The THz field propagates down the co-planar stripline and subsequently provides a transient bias field at the photoconductive switch. Transient currents measured across the detector photoconductive switch were used to extract the electric field between the co-planar stripline as a function of time delay between the photoexcitation of the sample and the THz readout. The detection system bandwidth, determined by the photoconductive switch and circuit response, limits sensitivity to frequencies below ~1 THz. This circuitry architecture enabled probing ultrafast electrodynamic properties without requiring ohmic contact to vdW samples, an active challenge for transition metal dichalcogenide materials[23]. All measurements were performed at 20 K.

### THz emission from directional photocurrent

We first observed THz emission from an edge-induced directional photocurrent in T$_d$-WTe$_2$. The heterostructure, denoted Device A, was photoexcited between the co-planar stripline traces, on the left and right edges of the heterostructure (green circles, inset Fig. 2a). The emitted THz signal recorded by the THz detector for each excitation spot is shown in Fig. 2b, with the corresponding spectral information found in the Supplementary Fig. 9.

While the time-domain traces from each location are similar and only exhibit changes in magnitude with fluence (see Supplementary Note 4), their amplitudes are inverted in sign. The integrated time-domain trace is proportional to the current orthogonal to the co-planar stripline (see Supplementary Note 5). The inverted time-domain signal thus originates from net currents that propagate in opposite directions in the WTe$_2$, as illustrated by the blue and red arrows in the inset of Fig. 2a.

The emitted THz field indicates the existence of a bias-free, directional photocurrent[12]. In bulk WTe$_2$, the presence of mirror planes along both the $a$ and $b$ crystal axes (see Fig. 1a) generally prohibits by symmetry the generation of a net in-plane photocurrent. The THz emission signals were maximized when the laser excited the edge of

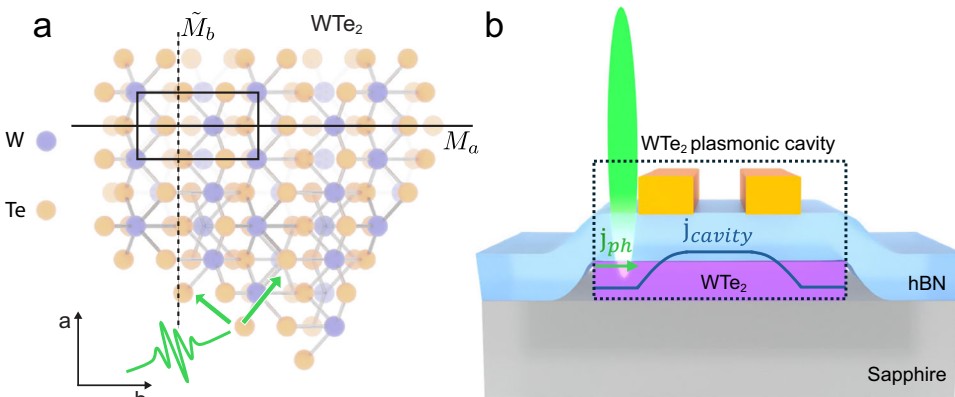

**Fig. 1 | Generation of edge directional photocurrents in T$_d$-WTe$_2$ enhanced by plasmonic modes of a self-cavity. a** Top view of the crystal structure of T$_d$-WTe$_2$ with the unit cell outlined and mirror planes of the $a$-axis ($M_a$) and the glide mirror plane along the $b$-axis ($\tilde{M}_b$) marked respectively. When photoexcited (green pulse) on an edge of the flake that is not parallel to a crystal axis, a net current can form. The green arrows indicate possible current directions, whose precise direction in each device is dependent on the edge orientation[10,28]. **b** Diagram of the cavity

device geometry. The co-planar stripline on top of the heterostructure guides the emitted THz to the detector switch, and modifies the dielectric environment, which, in conjunction with the edges of the flake, defines the plasmonic self-cavity. The current density of the cavity mode ($j_{cavity}$, in blue) changes significantly underneath the metal traces, due to screening. The directional photocurrent ($j_{ph}$, indicated by the green arrow), generated at the edges, is modified by the self-cavity feedback, enhancing emission at the cavity resonance frequency.

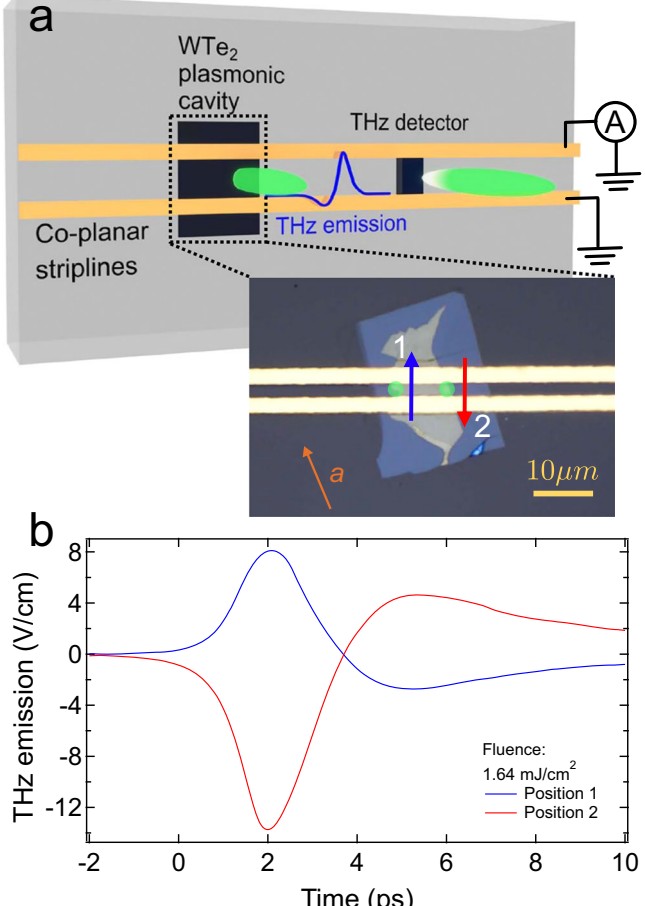

**Fig. 2 | Schematic of THz emission devices and measurement of photocurrents in WTe₂. a** Diagram of a heterostructure with THz optoelectronic circuitry. The inset shows Device A, a vdW heterostructure consisting of a 10 nm thick WTe₂ flake (yellow) and a 26 nm thick hBN insulating layer (blue), with a gold co-planar stripline passing over the top. hBN electrically insulated the WTe₂, such that the emission signal was capacitively coupled into the co-planar stripline. When a photocurrent was generated at the edge of the WTe₂ flake, the emitted THz field, corresponding to the derivative of the current in time, was coupled into the co-planar stripline and detected as a function of the time delay between the excitation and detection laser pulses. **b** Time-domain emitted field traces after excitation at a fluence of 1.64 mJ/cm² on the left (blue) and right (red) edges of WTe₂, at the positions marked by green circles in the inset of panel (**a**). Arrows demonstrate the counter-propagating currents, as expected by symmetry[10].

the heterostructure, indicating the relevance of the edge-broken mirror-symmetry (See Supplementary Fig. 8). Several experiments have reported such directional photocurrents generated at the edge of semimetals using techniques ranging from DC transport to time-resolved optical measurements[10,24–29]. These photocurrents have been attributed both to quantum geometric responses related to the crystal symmetry (see Supplementary Note 1), such as shift or photogalvanic currents[10,30]), as well as photothermal effects due to anisotropies between the Seebeck coefficients of the $a$ and $b$ crystal axes[28,29]. The experimental conditions reported here cannot resolve the ongoing discussion about the microscopic mechanism of this current. We refer to this as a directional photocurrent and leave the question of its microscopic origin to future work.

This photocurrent decays on ~picosecond timescales, likely due to carrier-carrier, and subsequent carrier-phonon, scattering[31] (see Supplementary Note 5 for details). While the rectified current itself is expected to be maximized at zero frequency, the Fourier transform of

the time-domain signal peaks in the frequency domain centered around 0.1 THz (see Supplementary Fig. 6), due to the capacitive coupling to the co-planar stripline and spectral sensitivity of the readout switch.

## Purcell enhancement of directional photocurrents

The fluence-dependent THz emission signal was investigated under photoexcitation at other spatial locations on the heterostructure to interrogate the self-cavity modification to the directional photo-current. Intriguingly, when the laser was positioned on the lateral edge, but outside the co-planar stripline (Fig. 3a, inset), the dynamics of the emitted field were dramatically transformed. Compared with the emission trace of Fig. 2b, the time-domain dynamics exhibit a fluence-dependent distortion (Fig. 3a), which reflects the interference of different frequency components (see Supplementary Note 6 for analysis using a finite impulse response filter of the time domain data).

Real and imaginary frequency domain data of the detected electric field are shown in Fig. 3b. At low fluences, most of the spectral weight is found ~0.1 THz, consistent with the directional photocurrent response observed above. As the fluence was increased, a resonance appeared at ~0.4 THz, whose complex lineshape is reminiscent of a Drude–Lorentz model.

To characterize the behavior of this peak, the real part of the emitted fields was fit with a Lorentzian resonance (see Supplementary Fig. 5 for details). The peak amplitude and resonance frequency were extracted and plotted as a function of fluence in Fig. 3c.

Similar measurements were repeated on an additional three devices, each with different flake thicknesses and crystal axes alignment relative to the emission sensing direction (see Supplementary Fig. 3 for each heterostructure geometry). Data from Device B are shown in Fig. 3d–f, with results from C and D found in the Supplementary Fig. 12. While the low fluence time-domain dynamics for Device B are comparable to that of the photocurrent response in Fig. 2b, the emitted field flips sign at high fluence.

As seen in Fig. 3e, a finite-frequency resonance centered around ~0.25 THz is observed to grow in amplitude and shift to lower frequencies as a function of fluence. The real part of the Fourier transform data can be fit to extract the power-dependent amplitude for Device B, plotted in Fig. 3d. The linear scaling fluence indicates that the excitation mechanism is second-order in field.

We interpret the spatially- and orientationally-dependent dynamics, when combined with this second-order field scaling, as a Purcell enhanced directional photocurrent, a shift in spectral weight from low frequencies to a sharp resonance in the THz regime. The modified emission characteristic of the Purcell effect[15] is a signature of an engineered photonic density of states in a cavity. While the canonical Purcell effect describes changes to the spontaneous emission rate of a dipole placed inside a cavity, the vdW heterostructures described in this experiment serve as both the emitter and the cavity. The photocurrent excited in the heterostructures can be thought of as a sum of radiating dipoles, whose individual discrete energies broaden into a continuous band.

The cavity can be described as a plasmonic self-cavity[13,32–37]. The devices investigated here are ~10 μm in lateral size and 5–70 nm thick, orders of magnitude smaller than the diffraction limit of THz frequency light. The edges of the flake and the patterned dielectric environment provided by the presence of the co-planar stripline traces impose boundary conditions on the plasmonic light–matter modes that can be sustained within the structure. These interfaces reflect and transmit currents, resulting in discretized standing wave patterns of current density.

To model these self-cavity modes and emission enhancement, we developed an analytical theory of the current density distribution inside the heterostructure after excitation by a rectified current, $j_{ph}$. We assumed that the edge current is first excited in the region outside

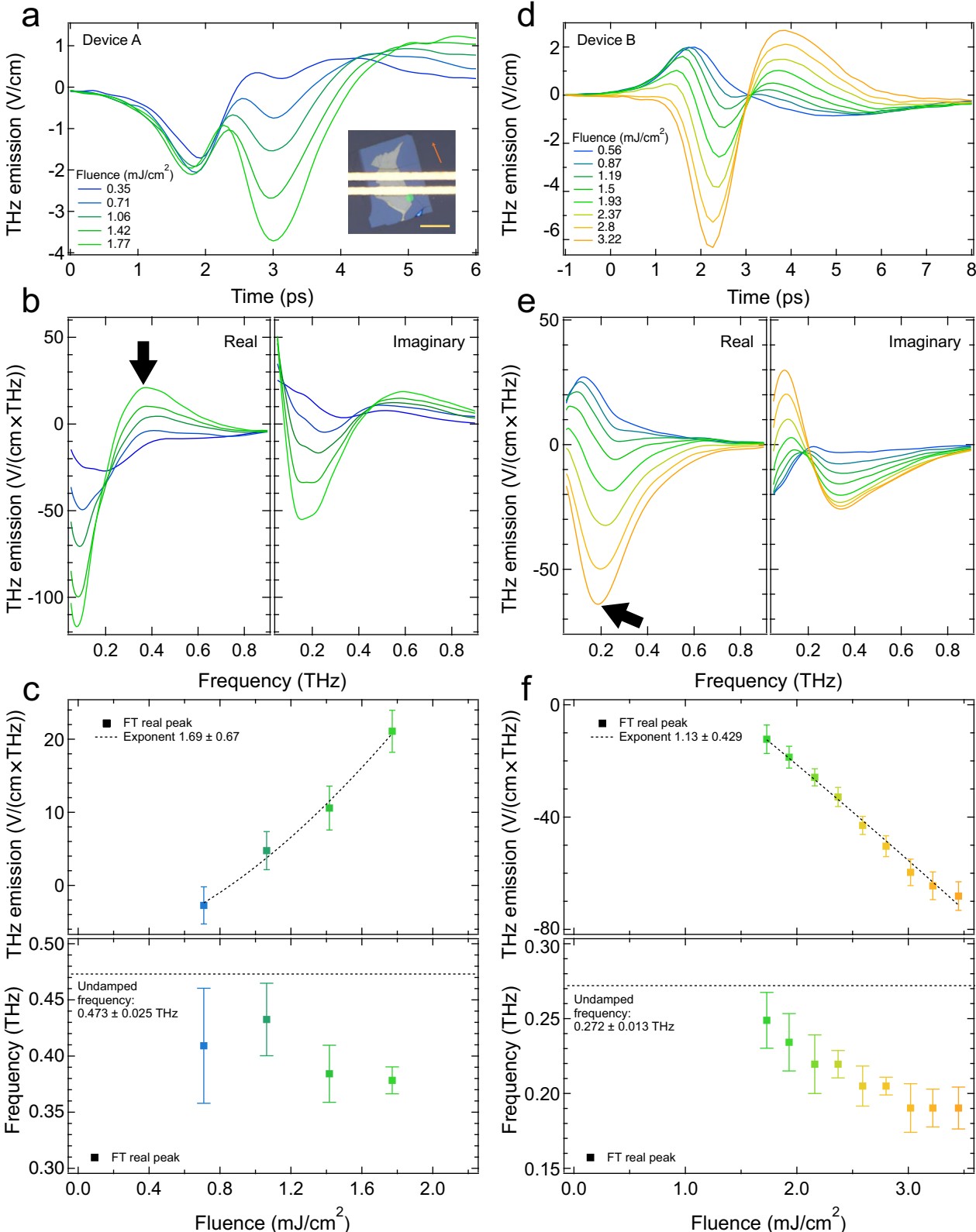

**Fig. 3 | THz emission from Purcell-enhanced directional photocurrent in WTe₂ self-cavities. a** The time-domain THz emitted field of Device A when photoexcited outside the co-planar stripline, as shown in the micrograph inset. The orange arrow illustrates the direction of the crystalline *a*-axis. The yellow bar indicates 10 μm. **b** Fourier transform of the time-domain field from panel (**a**), with the finite frequency resonance, corresponding to the Purcell enhancement, indicated by a black arrow. **c** Fluence dependence of the THz emission peak extracted from (**b**) (top)

and resonance frequency (bottom), used in conjunction with the cavity quality factor to calculate the undamped frequency (dashed line). Error bars indicate the uncertainty of the peak position (see Supplementary Note 3). **d** Time domain traces at varying fluences, **e** Fourier-transformed spectra showing resonance evolution, and **f** resonance amplitude and frequency for Device B. Dashed line indicates undamped frequency and error bars indicate the uncertainty of the peak position (see Supplementary Note 3).

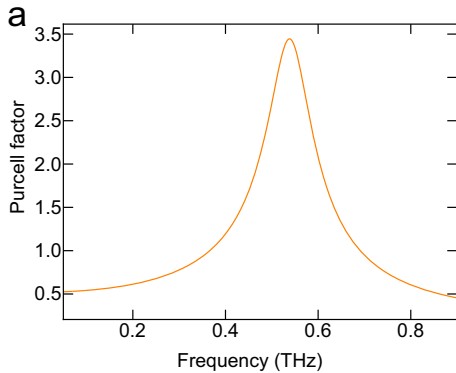
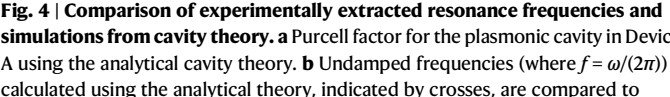
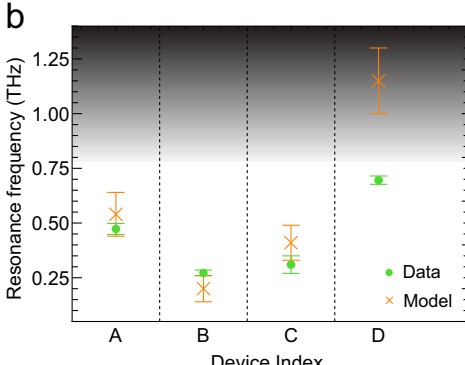

**Fig. 4 | Comparison of experimentally extracted resonance frequencies and simulations from cavity theory. a** Purcell factor for the plasmonic cavity in Device A using the analytical cavity theory. **b** Undamped frequencies (where $f = \omega/(2\pi)$) calculated using the analytical theory, indicated by crosses, are compared to experimental measurements, indicated by circles. The shaded region indicates frequencies outside of the experimental bandwidth. The uncertainty of the experimental data is discussed in Supplementary Note 3, and the uncertainty of the model is described in Supplementary Note 10.

of the co-planar strip, which subsequently propagates through the device, experiencing distinct dielectric environments, as well as different dissipation rates, $\gamma$, in regions underneath the co-planar strips versus outside or between these metal traces. The total current can be described as the sum of the transmitted and reflected currents, whose current density ($j(\omega)$) must fall to zero at the edges of the flake, and which, in addition to the potential ($V(\omega)$), must be continuous at the boundaries of the coplanar strip traces.

Using these boundary conditions, we solved Maxwell's equations for the current density and electric potential to determine the induced current density, $j_{cav}(\omega)$. As $j_{cav}(\omega)$ is linearly proportional to $j_{ph}(\omega)$ for all frequencies, this captures the feedback from the boundary conditions on the initial current, while $j_{ph}(\omega)$ determines the initial excitation bandwidth[13,37]. While the standing wave cavity modes that comprise $j_{cav}(\omega)$ span the entire device, the detected radiation predominantly originates from the current in the stripline gap (region 2, see Supplementary Fig. 13), labeled as $j_{cav,2}$ (see Supplementary Fig. 13), as this couples efficiently into propagating modes in the coplanar stripline, and is the current responsible for the detected THz. The detection of the resonance provides a read-out mechanism that is dependent on the response of the entire self-cavity mode.

Having developed the self-cavity model, we then define the angular frequency ($\omega$) dependent Purcell factor,

$$F(\omega) = \left| \frac{j_{cav,2}(\omega)}{j_{ph}(\omega)} \right|, \tag{1}$$

as the absolute value of the ratio of the current density ($j_{cav,2}$) in the co-planar stripline gap to the initial intrinsic photocurrent ($j_{ph}$). The Purcell factor can be used to identify frequencies where emission is expected to be maximized due to constructive interference of currents in the device, as a function of geometric design of the heterostructure, including the flake width, crystal axes alignment, hBN and WTe$_2$ thicknesses, as well as co-planar stripline dimensions. Further details of the analytical theory are found in the Supplementary Note 8.

Using this model, we simulated the Purcell factor for each device, with Device A shown in Fig. 4a; and additionally for Devices B–D in the Supplementary Fig. 14. The predicted peak aligns closely with the experimental resonances. We focus our discussion on the resonance frequency of each device, as the precise lineshape of the experimentally measured data corresponds to the directional photocurrent that is enhanced by the Purcell factor, and is thus dependent both on the initial photocurrent spectra and the detection switch response function, factors that are not captured in the Purcell factor theory.

To directly compare the theoretically calculated resonances to the experimental data, we must account for the experimentally observed softening in resonance frequency and broadened linewidth. These signatures are consistent with a damped harmonic oscillator, wherein the resonant frequency ($f_1$) is expected to redshift as dissipation increases. The expected undamped resonance frequency ($f_0$) can be extracted using the relation, $f_0 = f_1(1 - \frac{1}{4Q^2})^{-1/2}$, where the cavity quality factor, $Q$, is determined by $Q = \frac{2\pi f_1}{\gamma}$[38]. $\gamma$ is related to the linewidth of the resonance, attributed in the experiments to non-radiative dissipation routes, like fluence-dependent electron-electron interactions and electron-phonon scattering[39,40]. Using this formula, the undamped resonance frequency was extracted from the experimental data, and indicated by the dashed lines in Fig. 3c, f, and circular markers in Fig. 4b.

In Fig. 4b, we compare these undamped resonance frequencies (circular markers)– calculated using the damped harmonic oscillator model and corresponding to the dashed lines in Fig. 3c, f–with the resonance peaks (cross markers) from our analytical model. The Purcell resonance frequency is influenced by parameters including the thicknesses of the WTe$_2$ and hBN layers, as well as the width of the transmission line. Additional details are provided in Supplementary Notes 8 and 10. These points do not represent the peak positions observed in the raw data, but rather the peak positions after correcting for the damping effect described above. Error bars were calculated, including both statistical and systematic errors introduced by fitting and the frequency resolution of the experimental data.

The experimental undamped resonance frequency position agrees well with the analytical theory assuming no damping (see Supplementary Note 10). Deviations between experimental findings from devices C and D and the theory are attributed to the lack of high fluence data measured for device C, making it challenging to accurately extract the undamped position, and the substantially thicker WTe$_2$ flake of device D, which pushes the expected resonance frequency position outside of the experimental bandwidth and may require additional corrections to the theory due to greater screening. These results confirm that the observed THz emission resonances stem from Purcell-enhanced directional photocurrents, mediated by the modified density of states from plasmonic self-cavity modes.

## Discussion

The WTe$_2$ heterostructures exhibit a directional photocurrent upon photoexcitation at a flake edge. However, rather than an emission profile expected based on the time derivative of a rectified current, the emitted field exhibits a resonance at hundreds of gigahertz.

We understand the resonance in the emission spectrum as originating from a near-field Purcell effect. This process can be framed in terms of Fermi's Golden Rule[15],

$$\Gamma_{j_{ph} \rightarrow j_{cav}} \propto |\langle j_{cav}(\omega)|H'|j_{ph}(\omega)\rangle|^2 \rho(\omega), \qquad (2)$$

to describe the transition rate, ($\Gamma$), from the initial current density, ($j_{ph}$), to the current density of the cavity, ($j_{cav}(\omega)$), where $H'$ is a Hamiltonian coupling current into the self-cavity region of the device, and $\rho(\omega)$ captures the density of states of the cavity mode. In the absence of cavity effects, the plasmonic density of states is featureless. However, the boundary conditions of the self-cavity modify $\rho(\omega)$, such that the plasmonic density of states is proportional to the standing waves of current density described by the Purcell factor.

When the photocurrent spectrum is projected onto this plasmonic density of states, the result is an enhancement of the current spectrum resonant with the self-cavity modes, and suppression of emission at non-resonant frequencies (see Fig. 4a). The enhancement we define here is a frequency-dependent enhancement, specifically, the appearance of a new resonance peak due to the plasmonic cavity. If the sample were a large 2D flake without the patterned dielectric environment introduced by the stripline, the frequency spectrum would be trivial, exhibiting only a single photocurrent peak. In contrast, the geometry of the structure and the flake can be engineered so as to enhance or suppress specific frequencies relative to a flake without the stripline patterning. Effectively, photoexcitation generates a photocurrent response with frequency-dependent current density peaked at zero frequency. Interference of currents reflected at the boundaries shifts the spectral weight of the directional photocurrent to the resonance frequency of the cavity, which can be thought of as multiplying the initial photocurrent density spectrum by the Purcell factor. The experimentally detected THz emission spectrum is further convolved with the response function of the detection switch. As $\rho(\omega)$ and thus the Purcell factors are strongly geometry-dependent, the frequency and intensity of the resonantly-enhanced THz emission can be tuned via the device geometry.

The suppression, or destructive interference, of currents is most evident in Device B, where the spectral weight of the low-frequency directional photocurrent is shifted to the higher frequency resonance of the cavity mode. In particular, a phase difference is observed between the directional photocurrent, dominant at low fluences, and the cavity-mode, dominant at high fluences, which results in a sign reversal of the emitted THz field.

At high fluence, Device A shows both the DC photocurrent and Purcell resonance, whereas in Device B, only the Purcell contribution is detectable. This occurs because the Purcell peak in Device B is at a much lower frequency ($\approx$0.25 THz), causing the nearby spectral weight to shift from the DC photocurrent into the Purcell mode. The DC component in Device B is not physically absent, but lies below our 50 GHz detection limit and is therefore not observed[29].

The Purcell-enhanced emission is notably not observed when the heterostructure is photoexcited between the stripline traces, such as in Fig. 2. This lack of enhancement is due to the increased scattering that occurs within the detection region when the device is photoexcited between the stripline, resulting in an overdamped cavity mode. The emission spectra are determined by the boundary conditions the current experiences when propagating from one region of the flake to another. Direct excitation with the pump laser can induce local heating in the illuminated area. When direct excitation is in between the metal strips, this drives the cavity into the overdamped regime; when the excitation is outside the stripline, the heating effect on the detected photocurrents is much less pronounced. Thus, when exciting between the stripline, only the photocurrent component is present, whereas when exciting outside the stripline, the Purcell-enhancement can be detected in addition to the photocurrent (see Fig. 2 and Fig. 3a for

Device A). Furthermore, Supplementary Note 11 provides a detailed model describing this effect.

It is important to highlight that previous ultrafast studies of WTe$_2$ observed a photoinduced structural phase transition in WTe$_2$[41–43]. Despite comparable excitation fluences, no evidence of a photoinduced phase transition, or the interlayer shear phonon at ~0.2 THz suggested to drive the transition, was observed throughout this study. However, the fluence-dependent emission spectra exhibit nonlinear trends. While the model used is both linear and static, as $j_{cav} \propto j_{ph}$, the nonlinearities suggest the need for future theoretical work on fluence-dependent Purcell enhancement. For instance, the nonlinearities could occur due to the formation of Floquet bands, which modify dissipation channels, or nonlinear effects due to the formation of plasmons, like self-focusing or self-amplification. The notable absence of the phase transition also raises the possibility that light-matter coupling could suppress such phase transitions. The enhanced emission at the cavity resonances could be wielded to engineer dissipation or coupling between internal degrees of freedom, providing a route for controlling light-induced states. Future work could examine these effects in 2D and gate-tunable systems, where one layer of a heterostructure can be used as a cavity to engineer the responses in the other layer[13].

Our findings demonstrate a cavity-tunable, coherent, narrowband THz emitter in a challenging-to-access frequency range. By selecting the fluence and excitation position, one can tune the frequency of emission from DC to finite frequencies. In addition, the resonance frequency is tunable by sample geometry and can be predicted with good accuracy using the analytic theory developed here. The observed THz generation efficiency per thickness of 0.5 V/(cm nm) for WTe$_2$ exceeds the 0.39 V/(cm nm) for the amorphous silicon used in this study as a detector photoconductive switch (Supplementary Note 13), and the efficiency found for THz emitters ZnTe ($3.5 \times 10^{-3}$ V/(cm nm)) and vdW material NbOI$_2$ ($14 \times 10^{-3}$ V/(cm nm))[44]. This approach may provide a route to use plasmonic self-cavities as bias-free THz emitters for spectroscopy, wireless communication, or on-chip signal generation in frequency bands challenging to access with conventional electronics.

## Methods

### Device fabrication

WTe$_2$ flakes were exfoliated using scotch tape from bulk crystals (from 2D Semiconductors for Devices B–D, and grown by C.H. and J.M.D., under supervision from X.X and J-H.C. for Device A). The surface and thickness of the flakes were characterized with an atomic force microscope. Using a dry-transfer method, we prepared a stamp by placing a small drop of polydimethylsiloxane onto a microscope slide and covering it with a layer of polypropylene carbonate. Under a light microscope, both WTe$_2$ and hBN were transferred using the stamp on a transfer station. Once the stack was built, the samples were spin-coated with LOR-7B and maP-1205, followed by optical lithography to write the circuit patterns. We developed the structure with the maD-331/S developer for 35 s. Subsequently, 165 nm Si was evaporated at a rate of 10 Å/s for the photoconductive switches, and subsequently a 10 nm Ti sticking layer and 275 nm of Au at 0.5 Å/s for the coplanar stripline. After evaporation, the samples were immersed in PG remover for at least 24 h.

### Experimental setup

Optical pulses were generated using a 1030 nm laser with a FWHM pulse duration of ~100 fs at a repetition rate of 200 kHz, whose output was frequency doubled using a BBO crystal. The second harmonic output was split into two paths, and a mechanical delay was added to generate a time delay between the excitation and signal readout lines. A mechanical chopper was used to modulate the heterostructure excitation beam. The excitation beam was focused onto the heterostructure, and the readout onto the detection switch. Currents were

measured using a home-built transimpedance amplifier, and measurements were demodulated using a lock-in amplifier at the frequency of a mechanical chopper placed in the excitation path. All measurements took place at 20 K with the sample in an ARS cryostat.

## Data availability

The datasets generated and/or analyzed during the current study are available in the Edmond repository, at https://doi.org/10.17617/3.GRRCHN.

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

## Acknowledgements

We thank Sambuddha Chattopadhyay for helpful discussions, and the support of Boris Fiedler, Birger Köhling, Elena König, Toru Matsuyama, and Guido Meier for fabrication of electronic devices and support in the clean room. Funding: Device fabrication, terahertz optoelectronic measurements, theoretical development, numerical simulations, data analysis, and manuscript preparation were supported by the U.S. Department of Energy, Office of Science, Basic Energy Sciences, under Early Career Award DE-SC0024334. The growth and characterization of

WTe$_2$ single crystals and second harmonic characterization were supported by the Center on Programmable Quantum Materials, an Energy Frontier Research Center funded by the U.S. Department of Energy (DOE), Office of Science, Basic Energy Sciences (BES), under award DE-SC0019443. J.M.D was supported by the National Science Foundation Graduate Research Fellowship Program under Grant No. DGE-2140004. Any opinions, findings, and conclusions or recommendations expressed in this material are those of the authors and do not necessarily reflect the views of the National Science Foundation. The Max Planck-New York Center for Non-Equilibrium Quantum Phenomena supported experimental infrastructure. The growth of high-quality hBN performed by K.W. and T.T. was supported by the JSPS KAKENHI (Grant Numbers 21H05233 and 23H02052), the CREST (JPMJCR24A5), JST, and World Premier International Research Center Initiative (WPI), MEXT, Japan.

## Author contributions

J.W.M. conceived the idea for the experiment and supervised the overall project with assistance from H.M.B. On-chip THz devices were fabricated by X.L. and J.H. THz emission measurements and data analysis were performed by X.L. and J.H. with assistance from H.M.B, M.W.D., and B.S. Second harmonic generation measurements were performed by K.K. and F.S. with support from Y.H., V.Q.C., C.T., and Z.H.P. under the supervision of P.J.S and X.Y.Z. Theory was developed by M.H.M. with support from G.K., A.M.P, X.L., J.S., and M.F. under the supervision of A.R. Single-crystal WTe$_2$ samples for Device A were grown by C.H. and J.M.D., under supervision from X.X and J-H.C. High-quality hBN was grown by K.W. and T.T. The manuscript was written by X.L., J.H., M.W.D., H.M.B., and J.W.M. with contributions from all authors.

## Funding

## Competing interests

The authors declare no competing interests.

## Additional information

$^1$Max Planck Institute for the Structure and Dynamics of Matter, Hamburg, Germany. $^2$Center for Free-Electron Laser Science, Hamburg, Germany. $^3$Department of Physics, Columbia University, New York, NY, USA. $^4$Department of Chemistry, Columbia University, New York, NY, USA. $^5$Department of Mechanical Engineering, Columbia University, New York, NY, USA. $^6$Physics Department, Politecnico di Milano, Milan, Italy. $^7$Department of Physics, University of Washington, Seattle, WA, USA. $^8$Research Center for Materials Nanoarchitectonics, National Institute for Materials Science, Tsukuba, Japan. $^9$Research Center for Electronic and Optical Materials, National Institute for Materials Science, Tsukuba, Japan. $^{10}$Initiative for Computational Catalysis, Simons Foundation Flatiron Institute, New York, USA. ✉e-mail: hope.bretscher@mpsd.mpg.de; jm5382@columbia.edu

