## [Transparent Peer Review file · Nature Communications]

Purcell enhancement of directional edge photocurrent in a van der Waals self-cavity

Corresponding Author: Professor James McIver

Version 0:

Reviewer comments:

Reviewer #1

(Remarks to the Author)

The manuscript reports a self-cavity-induced Purcell enhancement of photogalvanic currents in the van der Waals semimetal WTe_2 . Using ultrafast optoelectronic circuitry, the authors present measurements of fluence-dependent nonlinear photocurrents excited at the sample edges. The dataset is extensive and of high quality. The authors attribute the observed effects to Purcell enhancement governed by self-formed cavity boundaries. The technique is powerful, and the experimental execution is commendable.

This referee agrees that self-cavity engineering offers a promising and novel approach to controlling nonlinear, nonequilibrium dynamics in quantum materials. However, there are several issues that must be addressed before the physical interpretation can be convincingly established. At this stage, I cannot recommend publication in its current form and encourage the authors to carefully consider the following points:

1. The DC photocurrent spectra appear significantly smaller than the cavity modes, raising questions about whether off-resonant coupling into the cavity is efficient. Why should this process be effective if the excitation is not resonant with the cavity mode? One would expect much more efficient coupling—and thus a stronger signal—when the excitation field spatially overlaps with cavity mode. Even if spatial sampling is a limiting factor, some fluence dependence in the lineshape should still appear. However, such fluence dependence inside cavity is not evident in Figure S7 or Figure 2.

General Interpretation Issues:

While the data quality is high and the study explores an interesting physical phenomenon, the interpretation is intricate and sometimes speculative. Several spectral features remain unexplained or are inconsistently discussed. These unresolved issues are critical, especially because they form the basis for differentiating this work from prior studies on similar materials. Figure 2: Clarification is needed on how excitation at central (non-edge) regions compares to edge excitation. Can the authors differentiate bulk versus edge contributions to the photogalvanic effect?

Figure 3: The large device-to-device variation is concerning. For example, Device A and Device B show opposite signs in their photocurrent peaks. If this is attributed to a phase effect, then why not report the amplitude or power spectra which potentially remove the misleading factors? Additionally, why does the DC current in Device B flip in phase vs the cavity resonance?

Moreover, at low fluence, Device A begins to resemble Device B at high fluence in showing a negative peak near 0.2 THz, with a redshift toward zero frequency as fluence increases. This suggests possible underlying common mechanisms not fully addressed in the interpretation.

The comparison between excitation inside and outside the cavity suggests real effects, but the interpretation is complex and highly dependent on excitation geometry. It would greatly strengthen the argument to include a systematic spatial mapping of excitation spots—especially further moving away from the coplanar waveguide. If the interpretation is correct, one should observe some correlations where the threshold power for generating nonlinear photocurrents decreases with distance from the cavity.

Figure 4:

It is unclear which data directly demonstrates the peak Purcell factor enhancement near 0.5 THz in Figure 4a. From Figures 3e and 3f, the signals actually appear stronger when shifted away from the 0.5 THz peak, which raises questions about the claim. A clear and fair comparison under matched excitation conditions is needed to convincingly support the enhancement shown in Figure 4a.

In summary, this manuscript presents intriguing and potentially impactful results: using self-formed cavity boundaries in quantum materials to manipulate nonlinear photocurrents via Purcell enhancement. However, the physical interpretation is intricate and, at times, not fully supported by the presented data. Several key spectral features remain unexplained or inconsistently addressed. Given the novelty of the interpretation and its dependence on subtle spectral signatures, these ambiguities must be resolved before publication. I encourage the authors to address the concerns above with more analysis, clarification of the physical mechanisms with potentially additional measurements.

Reviewer #2

(Remarks to the Author)

The manuscript entitled "Purcell enhancement of photogalvanic currents in a van der Waals plasmonic self-cavity" by Li et al describe cavities can enhance light-matter interactions in quantum materials by modifying optical and electronic responses at specific frequencies. Exfoliated van der Waals materials like WTe₂ can form intrinsic self-cavities that confine electromagnetic fields into plasmonic modes. This study observes cavity-enhanced photogalvanic currents in WTe₂ through THz emission measurements. The emission frequency is tunable via excitation fluence and sample geometry due to plasmonic interference. The authors proposed an analytical model to quantify the cavity resonance and spectral behavior across devices. Overall, the experimental observation and the suggested concept of "Purcell effect for THz emission" are quite remarkable. However, there are several comments we must point out.

1. The author should provide the specific parameters of four devices. For example, the thickness of the WTe₂ sample obtained by AFM characterization, and the thickness of the materials used in the terahertz readout structure.
2. In the generation mechanism of terahertz waves, the author advocates that the main mechanism is the PGE (photovoltaic effect), but does not provide sufficient experimental evidence to rule out other possible mechanisms, such as the photon drag effect, the photothermal effect, the photo Dember effect, or the optical rectification effect. Although a preliminary analysis of these mechanisms is presented in the third paragraph on page 6 of the paper, further experimental validation is required. Especially when ruling out the photothermal effect, the judgment criteria of high-temperature conditions and time scale are debatable. First of all, the laser pumping itself can cause the thermal effect of the sample, leading to an increase in the lattice temperature. Secondly, existing literature has pointed out that the WTe₂ material also exhibits the photothermal effect at low temperatures, and it is just more significant at high temperatures [Nat. Commun 2022, 13 (1), 3909]. Therefore, its occurrence at low temperatures should not be ignored. In addition, the difference in the time scale may not stem from the characteristics of the mechanism itself, but rather be limited by the sensitivity and time resolution of the detection system [Nat. Phys 2023, 19 (4), 507-514 cited by the author]. Therefore, the conclusion of ruling out the photothermal effect based solely on the existing analysis is not sufficient.
3. In the position-dependent THz emission, is it sufficient to provide only the signals at the edge positions? Are there signals at other positions? Or is the signal minimum at the center and gradually increases as the position moves towards both sides? To better understand the radiation mechanism, it is recommended to supplement relevant data or diagrams in the attachment.
4. The manuscript does not provide a clear explanation for the observed sign reversal of the THz field in device B under high pump fluence, as depicted in Figure 3. Or to be more specifically and deeply, why should the Purcell enhancement (or equivalently the resonant emission) only happen and be observed for high pump fluences? In our understanding, the Purcell enhancement effect only modulate the photonic/plasmonic density of state so to alter the emission rate for those affected frequencies. And Purcell effect is intrinsically a linear effect, which means spectral modes are independent and the same enhanced emission spectra should be observed, rather than the power dependent enhancement observed in this manuscript. Therefore, the sign reversal and the nonlinear behavior for high excitation fluences indicate that the terahertz radiation signal in this work could be the result of complex combined action of multiple physical mechanisms, and current explanations are insufficient. Based on the analytical cavity model that the authors proposed, is it possible to simulate the flipping trend of the polarity of the terahertz radiation field under the dependence of the pump fluence, so as to provide support for the experimental results?
5. The Purcell factor is the strongest in device A in the manuscript, but it seems that the THz wave is not the strongest? What's the reason for this result? It is suggested that tables or graphs be provided in the attachment for comparison.
6. The Purcell-enhanced emission is not observed when the heterostructure is photoexcited between the stripline traces. This observation indicates that the resonance standing-wave model proposed by the authors may be incomplete to describe the whole experimental results. In the model, the authors "assume that the local photocurrent excitation rapidly propagates throughout the device", so we would expect that if the authors take the coefficients in the simulation of the case of "photoexcited outside the striplines" to simulate the case of "photoexcited inside the striplines", the authors should also get some Purcell-enhancement result similar to the Purcell factor and the resonance frequency shown in Fig. 4. The fact that Purcell enhanced emission is not observed in this case, indicates that the model is incomplete or flawed. By the way, we

noticed that in the equation of (23), there are three terms to sum to describe the cavity current j_{cav} : the right-going (reflected) current, the left-going (transmitted) current, and the intrinsic photocurrent itself. Whether the intrinsic photocurrent should still be counted here for the cavity current? Is there a double-count problem? The authors should clarify that.

Considering the above points, we believe that this manuscript is not suitable for publication in Nature Communications before being revised.

Reviewer #3

(Remarks to the Author)

Reviewer #4

(Remarks to the Author)

The manuscript investigates photocurrent generation in a layered 2D van der Waals self-cavity. A photocurrent is observed via ultrafast optoelectronic circuitry when light is shone at the edge of the sample where mirror symmetry is broken. An enhancement of the photocurrent is reported when light is shone outside the cavity, which the authors attribute to Purcell effects.

From a technological standpoint, the authors employ ultrafast optoelectronic circuitry that avoids issues associated with voltage biases and can reliably detect electric signals at terahertz frequencies. This is quite an instrumental and experimental feat: a technique first pioneered by some of the authors (e.g., in Ref. 18). The authors have now extended this technique to WTe₂. Overall, the manuscript is well-written, supported by detailed analysis, and presents promising applications in next-generation THz technologies that could appeal to a broad audience. I really like it. However, we do have some comments that we hope the authors elaborate on:

1. One major question I have is: to what extent is this truly an enhancement? This ties directly to the authors' claim that the photocurrent excited inside the cavity does not experience such enhancement (why is that?). For instance, if I compare the red line in Fig. 2b (or S5) with Fig. 3a, at a similar fluence of 1.64 and 1.77 mJ/cm² respectively, the only difference is that, in the former case, light shines between the striplines, while in the latter, it shines outside them. Yet, the THz emission peak in the former case is about 3-4 times larger than in the latter. If, as the authors claimed, this is an enhancement, shouldn't I expect the former to be smaller? Or could it be that when light shines inside the striplines, the enhancement is still there, but the frequency of the peak of Purcell factors is somewhat suppressed? And if so, why is that? A clear and simple explanation of what the simple experimental "smoking gun" of the Purcell enhancement is would be very nice, especially given this seeming difference between no Purcell enhancement inside the striplines, but only there outside the striplines. While I understand that a detailed explanation might be outside the scope of this current work, it leaves a rather unsettling taste that it is not clear what the signature is.

2. A little confusion in Fig. 3: device A exhibits a peak near ~0.1 THz (attributed to DC photocurrent) that is larger in amplitude than the Purcell-enhanced peak at ~0.4 THz, whereas this "DC peak" is absent in device B, making the latter appear more ideal. Could this difference be explained in terms of factors such as sample edge alignment, sample thickness, or the size of the light beam? In fact, how do the authors identify which peaks experience the Purcell enhancement: these seemed to be different features for device A and device B. To that point, what extent is the observed enhancement engineerable? Could the authors provide some qualitative insights, based on experimental trends or theoretical considerations, on how the frequency and amplitude of the enhancement might be tuned?

3. The authors say that the signal is generated by a photogalvanic current (Sec. 2.2) - I take this to mean a current that is generated not from a p-n junction origin (the authors later suggest that it is from shift current in Sec. 3). Given the importance of the striplines in the experimental set-up, is there an easy way of understanding how the striplines do not contribute to p-n junction-like photocurrent? Saying this clearly will really help the untrained reader zero in on how to say something is of p-n junction origin and what isn't.

Reviewer #5

(Remarks to the Author)

Version 1:

Reviewer comments:

Reviewer #1

(Remarks to the Author)

The authors have performed additional measurements and provided clarifications addressing my questions, which I appreciate. While some aspects of the interpretation do not yet fully align with the current Purcell picture, the experimental observations are robust. I therefore support publication of the revised manuscript.

Reviewer #2

(Remarks to the Author)

In the revised manuscript, some improvements have been made to address some of my questions. However, I am still hesitant to recommend its publication in Nature Communications. The detailed reasons are listed below:

(1)Based on the thickness of devices characterized by AFM, it is evident that there is a substantial difference in the thickness of WTe₂ among these devices. In particular, for device D, the thickness difference is nearly an order of magnitude. Wouldn't this difference in thickness have an impact on the experimental results? The thickness of WTe₂ is included in the model, which makes it highly questionable whether thickness affects the overall experimental results or not. Moreover, according to the Purcell factor simulation results of device D in Figs. S13 and S14 in the supplementary materials, it is evident that the simulation results of device D differ significantly from those of other devices.

(2)Although the author downplays the mechanism of terahertz emissions and attributes it to the directional edge photocurrent, the thermal effect has not been explicitly excluded. However, the theoretical derivation in the supplementary material still clearly uses the PGE effect for explanation. Moreover, the representation of this nonlinear susceptibility tensor appears to be incorrect, and the author should carry out a detailed examination.

(3) The photocurrent term ought to incorporate the time term, and the rate of time decay is associated with the terahertz spectrum. Nevertheless, the model indicates that the photocurrent term excited by the light pulse remains a constant value and does not vary with time. Then, is this simplified portrayal of the actual transient photocurrent truly reasonable?

I noted that other reviewers have also raised similar questions. They noted that while the experimental results presented in this manuscript are interesting, some theoretical explanations are confusing and unpersuasive. I share the reviewers' skepticism toward the speculative mechanistic explanations proposed by the authors.

Reviewer #3

(Remarks to the Author)

Reviewer #4

(Remarks to the Author)

The authors have made extensive efforts to answer the questions raised by the referees. The responses seem reasonable and I am happy to recommend the onward publication of the paper.

Reviewer #5

(Remarks to the Author)

Version 2:

Reviewer comments:

Reviewer #2

(Remarks to the Author)

The authors have addressed all my previous concerns. The revised manuscript can be published without further revision.

Reviewer #3

(Remarks to the Author)

Authors Response to Reviews

December 18, 2025

Reviewer 1

The manuscript reports a self-cavity-induced Purcell enhancement of photogalvanic currents in the van der Waals semimetal WTe_2 . Using ultrafast optoelectronic circuitry, the authors present measurements of fluence-dependent nonlinear photocurrents excited at the sample edges. The dataset is extensive and of high quality. The authors attribute the observed effects to Purcell enhancement governed by self-formed cavity boundaries. The technique is powerful, and the experimental execution is commendable.

This referee agrees that self-cavity engineering offers a promising and novel approach to controlling nonlinear, nonequilibrium dynamics in quantum materials. However, there are several issues that must be addressed before the physical interpretation can be convincingly established. At this stage, I cannot recommend publication in its current form and encourage the authors to carefully consider the following points:

#Response#: We thank the referee for the time and effort spent reviewing our manuscript and for their thoughtful, constructive comments. These have been valuable in improving the clarity of the work and presenting the novelty and impact more effectively. In response to the referee's questions and suggestions, we have made significant changes to both the main text and the supplementary information. All changes are highlighted in red in the revised manuscript for clarity. Below we address each comment point-by-point.

1. The DC photocurrent spectra appear significantly smaller than the cavity modes, raising questions about whether off-resonant coupling into the cavity is efficient. Why should this process be effective if the excitation is not resonant with the cavity mode? One would expect much more efficient coupling—and thus a stronger signal—when the excitation field spatially overlaps with cavity mode. Even if spatial sampling is a limiting factor, some fluence dependence in the lineshape should still appear. However, such fluence dependence inside cavity is not evident in Figure S7 or Figure 2.

General Interpretation Issues: While the data quality is high and the study explores an interesting physical phenomenon, the interpretation is intricate and sometimes speculative. Several spectral features remain unexplained or are inconsistently discussed. These unresolved issues are critical, especially because they form the basis for differentiating this work from prior studies on similar materials.

#Response#:

In the following response, we show that, in general, we have a unified picture to explain the dataset we gathered, and we also introduce further corrections to our previous analytical model to achieve a more uniform explanation. This work differs from previous studies in two key aspects:

Firstly, we directly excite the two-dimensional flake and detect a unique directional current at the edges without applying any bias voltage. Secondly, we observe a non-equilibrium effect as we increase the excitation

fluence, showing a change in the signal in the frequency domain as the fluence increases — an effect we refer to as the *Purcell enhancement*.

The referee raises two points in regard to the magnitude of the detected signal. In particular:

1. The impact of a difference in frequency between the cavity mode and DC photocurrent spectra on the detected THz signal.
2. The effect of the “spatial overlap”, between the location of photocurrent generation and detection.

1. Spectral component separation

The referee correctly observes that, in our devices, the dominant spectral component of the DC photocurrent lies at a frequency below our detection bandwidth (50 GHz), resulting in the measurement of a sideband at the lowest detectable frequency. Meanwhile, the cavity resonance varies from 0.2 THz to 0.8 THz across the four devices. We deliberately designed our devices to achieve specific resonance frequencies, using the Purcell resonance to shift the DC frequency component to the desired resonance frequency. Although the Purcell resonance can appear larger when the resonance is closer to DC, the enhancement ratio itself does not change.

In our case, the presence of the metal strips enables a **transfer in spectral weight** of the current density, which we describe via Fermi’s golden rule (Eq. 2 in the main text):

$$\Gamma_{j_{\text{ph}} \rightarrow j_{\text{cav}}} \propto |\langle j_{\text{cav}}(\omega) | H' | j_{\text{ph}}(\omega) \rangle|^2 \rho(\omega).$$

As shown in the Purcell factor simulation results in Fig. 4a, the resonance corresponds to the plasmon frequency of the plasmonic cavity. A Purcell factor greater than one means that the current density at the corresponding frequency is larger than it would be without the stripline present — in other words, it is the ratio between the frequency component in the cavity case and that in the non-cavity case.

Specifically, comparing the frequency components of the DC photocurrent in Fig. S5 with those after Purcell enhancement in Fig. 2b reveals the emergence of a new resonance peak in Fig. 2 — corresponding to an increased current density near the plasmonic resonance — which is absent in the pure DC photocurrent case. We refer to this frequency-specific increase in current density (and thus THz emission) as **Purcell enhancement**. Physically, this is not an amplification of the DC component, but a re-distribution of current density from frequencies where the Purcell factor is less than one (in our devices mainly near DC) towards the resonance frequency.

The lineshape and specific frequency dependent amplitude of the detected THz emission is determined both by the spectrum of the directional photocurrent, and how this is modified by the Purcell enhancement. The resonance peak amplitude can appear larger when the resonance is closer to DC, due to the higher current density originally present at that frequency; however, the enhancement ratio itself does not change. For example, in Fig. 3, Device B has a resonance frequency lower than that of Device A, so its photocurrent already has more spectral weight near resonance. Consequently, at higher fluence, the resonance mode dominates in Device B, while the THz frequency content is only weakly modified by the cavity. In contrast, Device A has significant photocurrent content near DC, and thus exhibits a clear new resonance peak at the cavity frequency — a clearer demonstration of the Purcell enhancement, capable of tailoring the spectral components and shifting current density to a different frequency.

2. Effect of spatial overlap (excitation between metal strips)

The referee notes that, in principle, stronger enhancement should occur when the optical excitation spatially overlaps more closely with the cavity structure — i.e. excitation between the metal strips. This

was indeed our initial approach, as shown in Fig. 2 and Fig. S7. However, in this configuration we did not observe an extra resonance peak; instead, only the DC photocurrent component was detectable.

We attribute this outcome to local heating induced by the pump laser. We previously proposed a qualitative explanation: direct excitation with the pump laser can cause local heating in the illuminated area, which increases the damping rate and drives the cavity into an overdamped regime when excited in the central region between strips. This heating effect is much less influential when the excitation is applied outside the stripline.

Our original analytical model in Section S.7 could describe excitation outside the stripline but did not fully explain why the Purcell resonance peak appears only in this geometry, not when exciting between metal strips.

To investigate this effect in greater detail, we extended our analytical model to include region-dependent damping coefficients, denoted γ_i .

Figure R1: **Cross section of the heterostructure with labeled regions.** The model assigns separate damping coefficients γ_i to each spatial region.

The geometry is divided into different regions (region 0 outside the stripline, region 2 inside the stripline gap). For areas not directly illuminated by the laser, we set $\gamma_i \equiv \gamma_{\text{cav}}$, corresponding to the experimentally extracted cavity damping. For areas directly subject to laser illumination, we compared two cases: 1. $\gamma_i = \gamma_{\text{cav}}$ (no additional heating effect), and 2. $\gamma_i = 50 \gamma_{\text{cav}}$ (representing a strong heating-induced increase in damping).

To estimate the possible effect of laser heating on the damping, we note that the optical constants of WTe_2 along the b -axis at 515 nm ($n \approx 3.2$, $k \approx 1.6$ [1]) give a reflectivity $R = \frac{(n-1)^2 + k^2}{(n+1)^2 + k^2} \approx 0.366$, corresponding to a worst-case absorbed fraction of $A_{\text{abs}} \approx 0.634$. The optical penetration depth is only $\delta \approx 5$ nm [2], so the absorbed energy density for a fluence F is $E_{\text{vol}} \approx FA_{\text{abs}}/\delta$. At low temperatures, the volumetric heat capacity follows the Debye law $C_{\text{vol}}(T) = \beta T^3$ with $\beta \approx 2.4 \times 10^{-5} \text{ J}/(\text{cm}^3 \text{ K}^4)$ [3, 4], giving the peak temperature after a single pulse as $T \approx \left(T_0^4 + \frac{4FA_{\text{abs}}}{\beta\delta}\right)^{1/4}$. For typical conditions ($T_0 = 20$ K, $F = 3$ mJ/cm²) this yields $T/T_0 \approx 8$.

In this material, the resistivity, ρ , scales as $\rho \propto T^2$ at low T [5]. The Drude model assumes that $\gamma \propto \rho$. Thus, one could expect that the linewidth will change as $\gamma/\gamma_0 \approx (T/T_0)^2 \approx 64$. This instantaneous heating thus could lead to an increase in damping on the order of 50, and thus we use in these calculations $\gamma \sim 50\gamma_0$ in the below estimates.

The calculated Purcell factors for Device A in all four cases are shown below:

Figure R2: **Simulated Purcell factors for Device A under various damping conditions.** Cases include excitation at region 0 (outside stripline) and region 2 (inside stripline gap), each evaluated with and without increased damping from local heating. The lower cut-off of the x-axis is 50 GHz, corresponding to the lower limit of our experimental bandwidth.

The simulation results show that when the laser excites region 0, the Purcell factor remains essentially unchanged in both low- and high-damping cases. This indicates that our detection is not strongly sensitive to extra damping outside the stripline gap.

In contrast, when the laser excites region 2 (inside the gap), the ideal case without additional damping still produces a Purcell enhancement of similar magnitude to the outside-excitation case. However, when heating increases the damping in region 2 ($\gamma_i = 50\gamma_{cav}$), the Purcell factor is strongly suppressed. Even with a more moderate damping enhancement of $\gamma_i = 8\gamma_{cav}$, the maximum Purcell factor remains only 1.1 when excitation occurs at the center, making it virtually undetectable in the raw data. This supports our previous qualitative picture: laser-induced heating in the gap lowers the cavity quality factor enough to push the system into an overdamped regime, eliminating the Purcell resonance peak. In this geometry, only the DC photocurrent component remains detectable, consistent with our experimental observations.

To better illustrate this mechanism, we provide a schematic timing diagram of the photocurrent generation and cavity interaction:

Figure R3: **Timing diagram of photocurrent generation.** (a) Excitation between metal strips: local heating (yellow region) significantly increases damping and pushes the cavity into the overdamped regime, scattering the photocurrent during propagation and suppressing Purcell enhancement (as in Fig. 2 and Fig. S7). (b) Excitation outside the stripline: local heating still occurs in the illuminated area but mainly in an off-centre region; a substantial fraction of the photocurrent propagates into the screened region under the metal strip, allowing reflection across the cavity with minimal quality-factor reduction. As a result, Purcell enhancement is preserved (consistent with Fig. 3).

This extended modeling quantitatively confirms our qualitative reasoning: the suppression of Purcell enhancement for excitation between strips can be explained by local heating-induced overdamping, whereas excitation outside the stripline maintains a sufficiently high cavity quality factor for the enhancement to be observed.

Figure 2: Clarification is needed on how excitation at central (non-edge) regions compares to edge excitation. Can the authors differentiate bulk versus edge contributions to the photogalvanic effect?

#Response#:

To address the reviewer’s question and specifically examine the distinction between photocurrent contributions originating from the edges versus the bulk, we performed additional position- and fluence-dependent measurements on Device A. This new dataset has been included in the Supplementary Material (Section S.4).

The measurements include a third excitation position located in the central (non-edge) region between the metal strips. The upper and lower panels in the figure below correspond to the same excitation positions as shown in the main-text Fig. 2, while the middle panel presents the new data at the central area.

Figure R4: **Comparison of photocurrent response for edge and central excitation in Device A.** Upper and lower panels (a, c): excitation positions as in Fig. 2 of the main text; middle panel (b): measurement at the central (non-edge) region between the metal strips, where the emitted signal is much weaker.

As described in the previous section (Fig. R3), when excitation occurs between the metal strips, any possible cavity-related response is strongly damped, leaving only the DC photocurrent component detectable. When exciting both between the metal strips and laterally in the middle of the flake, shown in the middle panel of Fig. R4, the photocurrent signal under comparable excitation fluence is significantly suppressed relative to that at the edge positions. Due to the limited spatial resolution of our setup (determined by the pump-beam size and the optomechanical stages and mirror that are used to position the laser on the flake) and possible effects arising from the presence of the striplines (such as the formation of symmetry breaking edge-like boundaries at the metal-strip edges), a perfectly zero signal is not observed. Nevertheless, the photocurrent is clearly maximized when the excitation position approaches the edges of the WTe_2 flakes.

Figure 3: The large device-to-device variation is concerning. For example, Device A and Device B show opposite signs in their photocurrent peaks. If this is attributed to a phase effect, then why not report the amplitude or power spectra which potentially remove the misleading factors? Additionally, why does the DC current in Device B flip in phase vs the cavity resonance? Moreover, at low fluence, Device A begins to resemble Device B at high fluence in showing a negative peak near 0.2 THz, with a redshift toward zero frequency as fluence increases. This suggests possible underlying common mechanisms not fully addressed in the interpretation.

The comparison between excitation inside and outside the cavity suggests real effects, but the interpretation is complex and highly dependent on excitation geometry. It would greatly strengthen the argument to

include a systematic spatial mapping of excitation spots—especially further moving away from the coplanar waveguide. If the interpretation is correct, one should observe some correlations where the threshold power for generating nonlinear photocurrents decreases with distance from the cavity.

#Response#:

We wish to emphasise that there is a consistency across datasets, captured by our model and interpretations. At low fluence, in every case the photocurrent component is the only dominant contribution. When the fluence is increased, additional dynamics appear when excitation is applied outside the stripline.

We intentionally designed, constructed, and measured devices that exhibit different Purcell resonance frequencies to underscore the tunability of these cavities. For instance, during the fabrication process, we intentionally selected the thicknesses of the hBN and WTe₂ flakes for Devices A and B to place their resonance frequencies at approximately 0.55 THz and 0.2 THz, respectively, as shown in Fig. 4b of the main text. Because of the differing frequencies of this additional mode, the behaviour in the time-domain traces appears different, but the peaks in the frequency domain convey the same underlying picture.

The referee raised concerns regarding details when comparing Device A and Device B, including:

1. Phase effect resulting in opposite signs of the real part of the FT in Device A and Device B.
2. Differing signs between the FT real part at DC frequencies versus the cavity resonance.
3. The use of presenting the amplitude versus real and imaginary parts of the emission data.
4. Comparison between Device A at low fluence and Device B at high fluence.

In addition, an additional question was raised regarding the spatial mapping of excitation position:

5. Systematic spatial mapping of excitation spots.

Below we explain in more depth the origins of variation in these signatures, as well as discussion of the spatial mapping and device tunability.

1. Phase effect for opposite sign in Device A and Device B

The referee is correct that there is a phase difference between devices. Specifically, in the experiment there exists an unknown $\pm\pi$ phase shift when measuring different devices. These phase differences occur due to the experimental methods. The data are collected by modulating the pump laser using a mechanical chopper wheel, which provides the clock cycle for a lock-in based detection scheme. However, the chopper position can vary during the experiment, so we can only ensure global phase consistency within the dataset recorded from a single device. Importantly, this $\pm\pi$ phase shift does not change the real or imaginary parts. Thus, the qualitative behavior remains consistent across devices, even though we cannot determine the precise sign of the propagation direction.

For Device A, the phase origin (0) is chosen for measurements at position 1 (left edge). Position 2 (right edge) exhibits currents that flow in the opposite direction. When exciting outside the stripline but still at the right edge, the direction of the DC component remains the same (compare Fig. 3a and Fig. 2b). For Device B, a larger excitation spot is used without position dependence; the phase origin is chosen at the lowest fluence for the photocurrent component. Thus, to compare between Device A and Device B, it is more appropriate to set the lowest fluence of both devices to phase 0, so that the photocurrent directions match; in this sense the two devices behave similarly.

2. Differing signs between the FT real part at DC frequencies versus the cavity resonance.

When ignoring the global π phase difference between Device A and Device B, it is apparent that the photocurrent peak near DC (dominant at low fluence) and the Purcell resonance peak at higher frequency (more obvious at high fluence) always have opposite signs in the frequency domain. This may be due to a phase delay associated with formation of the cavity resonance.

The time-domain data appears more complex and varies considerably across devices. The reason for this variation is the superposition of different frequency components. To gain a clearer understanding of the time-domain behaviour, one can apply a finite impulse response (FIR) filter. As an example for Device A, shown in Fig. R5, this analysis reveals a superposition of a single-cycle emission from the DC photocurrent component and a high-frequency oscillation from the Purcell resonance, which agrees well with the insights obtained from the frequency-domain data.

Figure R5: **Example of FIR filter applied to highest-fluence measurement of Device A.** (a, b) Time- and frequency-domain traces of Device A at 1.77 mJ/cm^2 , corresponding to the highest-fluence trace in Fig. 3 of the main text. (c, d) Time and frequency responses of the two FIR filters used; these select frequency components below 0.3 THz and between $0.3\text{--}1 \text{ THz}$, respectively.

A key difference between Device A and Device B is that, at high fluence, the photocurrent component remains pronounced in Device A but disappears in Device B, leaving only the Purcell contribution dominant. This is mainly because the Purcell resonance peak of Device B lies at a much lower frequency ($\approx 0.25 \text{ THz}$), so the existing spectral weight near that frequency is transferred from the DC photocurrent component to the Purcell component. This does not mean that the DC component is fully transferred in Device B. As shown in another similar measurement with a broader detection bandwidth, the photocurrent spectrum exhibits a Drude-type response, with its frequency components extending down to 10 GHz or lower [6]. However,

we cannot detect components below our 50 GHz lower limit. This makes the dataset appear as if only the Purcell resonance remains.

Applying a Fourier transformation to obtain the frequency-domain information allows us to better understand the data and analyse the emission components of each device. The emission components vary between devices due to deliberate differences in our design during fabrication, but this variation is not caused by sample inhomogeneity.

Across all datasets, the physical picture remains consistent: At low fluence, the traces in all devices are dominated by the photocurrent component. The photocurrent peak lies close to DC, and we detect its sideband within our measurement range (starting at 50 GHz). At high fluence, the photocurrent peak remains in most cases, while the Purcell-enhanced resonance peak becomes more pronounced.

A special case is Device B, where the proximity of its Purcell resonance peak to the low-frequency limit of our detection bandwidth affects the measurement of the photocurrent. As a result, at the highest fluence only the Purcell resonance peak is clearly distinguishable.

3. Amplitude versus real and imaginary of emission data

We use the real and imaginary parts (rather than the magnitude) to correspond to the energy loss (absorption) and phase effects, analogous to the complex conductivity of a Drude–Lorentz peak. The real part is then used to fit the resonance frequency and quality factor of the cavity.

This approach is both experimentally valid and required by our analytical model:

Experimentally, we have a well-defined global time zero, identified as the peak of the lowest-fluence emission. The only concern is that a $\pm\pi$ phase difference between devices may exist, which could cause an apparent sign flip. The overall sign is assigned arbitrarily based on the direction of the time trace at acquisition. However, this phase difference is not random and does not lead to an incorrect physical interpretation; therefore, we can still use the real and imaginary parts of the frequency-domain data for analysis.

From the analytical model, the resonance peak of our plasmonic cavity can be understood as a Drude–Lorentz peak. This inherently requires that the analysis separate the frequency-domain data into real and imaginary parts.

4. Comparison between Device A at low fluence and Device B at high fluence.

The signals appearing at low fluence in Device A and at high fluence in Device B thus occur due to different physical mechanisms and thus it is to be expected that they have different features or trends. For Device A the sideband of the photocurrent originates close to DC, whereas for Device B, the resonance peak arises from the LDOS transformed after Purcell enhancement.

The origin of these peaks is evidenced by both fluence dependent trends as well as comparisons to the analytical theory. First, we address the fluence dependence. In our experiments, at low fluence the photocurrent component is always dominant, whereas at high fluence an additional component becomes apparent, which we attribute to Purcell enhancement arising from a current-density transition. The sign reversal between these two modes, evident when comparing the frequency-domain data in Fig. 3b and Fig. 3e, also serves as supporting evidence for the existence of a new mode.

Secondly, the newly appeared dynamic matches the analytical model for all four devices. From the experimental data, we also observe a consistent redshift of the resonance peak with increasing fluence (Figs. 3c and 3f). We fit this behaviour using a damped-harmonic-oscillator model, which yields the *undamped* resonance frequency (shown as dashed lines in those panels). Using this undamped frequency as the ideal limit, we compare it with the Purcell resonance frequency calculated analytically for all four devices and find

good agreement (Fig. 4b). This agreement supports the identification of the additional mode as the Purcell resonance.

5. Systematic spatial mapping of excitation spots

Based on the model of local heating induced by the laser (Fig. R2), moving the pump beam further away from the stripline should yield a similar resonance, as the cavity damping is not significantly affected provided that the excitation occurs outside the metal strips. However, under experimental conditions, several complex factors can influence the final result. Fig. R6 shows additional position-dependent measurements we performed. The Purcell resonance at a distant position still exists but is less pronounced, due to a combination of effects including the altered excitation edge, dispersion of the photocurrent and complex boundary conditions.

In Fig. R6, top row corresponds to emission from a position similar to Fig. 3a, while the bottom dataset corresponds to excitation farther from the stripline. As shown in the fluence dependence in both the time and frequency domains, the photocurrent component decreases and the resonance peak becomes less pronounced as the excitation is moved farther away.

The reduction is hypothesized to occur due to two reasons.

Firstly, when the pump beam is moved far enough to produce a significant difference, the excitation position is ultimately located at a different crystal edge compared to the cases in Fig. 2 or Fig. 3 of the main text. In addition, the longer distance to the cavity introduces greater dispersion of the photocurrent. Taken together, these factors alter the photocurrent lineshape in Device A relative to that shown in Fig. 2 of the main text.

Secondly, excitation at edges far from the cavity structure means that the photocurrent experiences more nonuniform boundary conditions due to the irregular shape of the WTe_2 flake. We deliberately avoided this in our primary measurements by exciting closer to the stripline. Far-edge excitation causes more complex reflections and losses before the resonance builds up, thereby decreasing the formation and detectability of the Purcell peak.

This observation disfavors performing excitation far from the stripline in our experiments. In future work, these effects could be investigated after cutting or etching WTe_2 into well-defined shapes or squares, which would ensure excitation along the same edge while reducing sensitivity to irregular edge-scattering mechanisms.

Figure R6: **Extra position-dependent measurements for Device A.** Top panels (a): excitation near the cavity edge (similar to Fig. 3a). Bottom panels (b): excitation farther away from the cavity structure, showing reduced photocurrent and disappearance of the resonance peak.

This extra dataset has been added to Supplementary Section S.12.

Figure 4: It is unclear which data directly demonstrates the peak Purcell factor enhancement near 0.5 THz in Figure 4a. From Figures 3e and 3f, the signals actually appear stronger when shifted away from the 0.5 THz peak, which raises questions about the claim. A clear and fair comparison under matched excitation conditions is needed to convincingly support the enhancement shown in Figure 4a.

#Response#: The data shown in Fig. 3e/f do indeed correspond to the Purcell-enhanced emission signal, but comparing these directly to Fig. 4a requires an additional step to account for dissipation. In the experimental data, we observe that increased local heating leads to a redshift of the Purcell peak. This effect is not fully captured in our simplified analytical model.

To address this discrepancy, we fit the experimental data (of Device A) using a damped-harmonic-oscillator model and obtain the *undamped resonance frequency*, shown as dashed lines in the lower panels of Figs. 3c. We then compare this undamped resonance frequency with the peak predicted by the analytical model, as shown in Fig. 4a.

To answer this question in more depth, we further clarify two points:

1. Peak correspondence between Figs. 3b, 3e and Fig. 4.
2. Resonance peak shift with increasing fluence.

1. Peak correspondence between Figs. 3b, 3e and Fig. 4

The Purcell factor shown in Fig. 4a (main text) is a simulation result for Device A, corresponding to the Purcell resonance peak (appears at higher fluence) observed in Fig. 3b. In our structure, the Purcell resonance frequency depends on geometric factors and therefore differs for each device. Consequently, Fig. 4a

cannot be directly compared to the frequency-dependent spectra shown in Fig. 3e, which corresponds to Device B.

In Section S.9 (Fig. S11), we show simulated Purcell factors for the other three devices, using γ_{cav} , which represents conditions similar to those measured experimentally.

Below is a modified version of Fig. S11 for clarity:

Figure R7: **Purcell factors for all devices using experimental damping.** Panel a corresponds to Device A (same as Fig. 4a), while Panel b shows the case for Device B, with a Purcell resonance peak near 0.2 THz. Same for Device C and D.

In Fig. 4, the calculated Purcell peak for Device A is around 0.54 THz, whereas experimentally the Purcell peak at high fluence appears around 0.4 THz. This mismatch is explained by a resonance peak shift with increasing fluence, as described below.

2. Resonance peak shift with increasing fluence

From the experimental data, we observe a consistent redshift of the resonance peak as fluence increases (see Figs. 3c and 3f). We fit this behaviour using a damped-harmonic-oscillator model, which yields the undamped resonance frequency (shown as dashed lines in those panels). This undamped frequency corresponds to the ideal case of the plasmonic cavity. In our comparison with theory, we use this undamped frequency rather than the peak frequency at a given fluence.

In the analytical model, the damping is represented by a parameter γ . To illustrate the idealised situation, we calculated the Purcell factor in the limit $\gamma \rightarrow 0$, producing a well-defined resonance frequency:

Figure R8: **Purcell factors in the ideal ($\gamma \rightarrow 0$) case for all devices.** Represents undamped cavity resonances with high quality factors.

This $\gamma \rightarrow 0$ case assumes a cavity with a very high quality factor. Conversely, increasing γ in the model (poorer quality factor) also leads to a redshift of the resonance peak; however, the shift predicted by the simplified model is much smaller than that observed experimentally.

Therefore, to achieve a meaningful comparison between experiment and theory, Fig. 4b uses only the ideal-case limit: the undamped resonance frequencies extracted experimentally (dashed lines in Figs. 3c and 3f, lower panels) are compared with the resonance frequencies calculated in the $\gamma \rightarrow 0$ case of the analytical model.

Finally, in the theory the difference in resonance frequencies between the damped and undamped cases is relatively small; thus Fig. 4a remains a good representation of both the width and the resonance frequency of the Purcell mode.

In summary, this manuscript presents intriguing and potentially impactful results: using self-formed cavity boundaries in quantum materials to manipulate nonlinear photocurrents via Purcell enhancement. However, the physical interpretation is intricate and, at times, not fully supported by the presented data. Several key spectral features remain unexplained or inconsistently addressed. Given the novelty of the interpretation and its dependence on subtle spectral signatures, these ambiguities must be resolved before publication.

I encourage the authors to address the concerns above with more analysis, clarification of the physical mechanisms with potentially additional measurements.

#Response#: We thank the referee once again for the helpful points and suggestions. In response, we have made corrections and modifications to the manuscript, and have added additional data where appropriate.

Specifically, we have provided an extended part of the analytical model along with more detailed explanations of how to connect the theoretical framework to the experimental observations. Taken together, these improvements help to establish a unified understanding of the datasets we have gathered.

All changes are marked with text in red in the revised version. We will be happy to address any further comments or questions, and we look forward to the referee’s continued feedback.

References

- [1] Munkhbat, B., Wróbel, P., Antosiewicz, T. J. & Shegai, T. O. Optical constants of several multilayer transition metal dichalcogenides measured by spectroscopic ellipsometry in the 300–1700 nm range: high index, anisotropy, and hyperbolicity. *ACS photonics* **9**, 2398–2407 (2022).
- [2] Buchkov, K. *et al.* Anisotropic optical response of WTe₂ single crystals studied by ellipsometric analysis. *Nanomaterials* **11**, 2262 (2021).
- [3] Callanan, J. E., Hope, G., Weir, R. D. & Westrum Jr, E. F. Thermodynamic properties of tungsten ditelluride (WTe₂) i. the preparation and low temperature heat capacity at temperatures from 6 k to 326 k. *The Journal of Chemical Thermodynamics* **24**, 627–638 (1992).
- [4] Laboratory, Q. M. O. Tungsten ditelluride (wte₂) optical properties database (2024). URL <https://quantumlab.uark.edu/wte2/>. Accessed: 2024-07-06.
- [5] Perevalova, A. *et al.* Electronic transport in a topological semimetal WTe₂ single crystal. *arXiv preprint arXiv:2302.00297* (2023).
- [6] Chatterjee, S., Yoshioka, K., Wakamura, T., Perebeinos, V. & Kumada, N. Intrinsic ultrafast edge photocurrent dynamics in WTe₂ driven by broken crystal symmetry. *arXiv preprint arXiv:2510.06618* (2025).

Reviewer 2 & 3

The manuscript entitled “Purcell enhancement of photogalvanic currents in a van der Waals plasmonic self-cavity” by Li et al describe cavities can enhance light-matter interactions in quantum materials by modifying optical and electronic responses at specific frequencies. Exfoliated van der Waals materials like WTe₂ can form intrinsic self-cavities that confine electromagnetic fields into plasmonic modes. This study observes cavity-enhanced photogalvanic currents in WTe₂ through THz emission measurements. The emission frequency is tunable via excitation fluence and sample geometry due to plasmonic interference. The authors proposed an analytical model to quantify the cavity resonance and spectral behavior across devices. Overall, the experimental observation and the suggested concept of “Purcell effect for THz emission” are quite remarkable. However, there are several comments we must point out.

#Response#: We thank the referee for the time and effort spent reviewing our manuscript and for their thoughtful, constructive comments. These have been valuable in improving the clarity of the work and presenting the novelty and impact more effectively. In response to the referee’s questions and suggestions, we have made significant changes to both the main text and the supplementary information. All changes are highlighted in red in the revised manuscript for clarity. Below we address each comment point-by-point.

1. The author should provide the specific parameters of four devices. For example, the thickness of the WTe₂ sample obtained by AFM characterization, and the thickness of the materials used in the terahertz readout structure.

#Response#: The thicknesses shown in Fig. S2 correspond to the individual flakes measured by AFM. For clarity, we summarise the parameters in the table below and have added this table to Section S.2 of the Supplementary Materials.

Table 1: Thickness parameters for Devices A–D. The metal strip consists of 10 nm Ti and 275 nm Au, deposited under identical evaporation conditions, and the amorphous silicon (a-Si) photoconductive switch has a constant thickness of 165 nm.

Device	hBN thickness (nm)	WTe ₂ thickness (nm)	Additional layers
A	26	10	10 nm Ti + 275 nm Au & 165 nm a-Si
B	8.3	6.8	
C	16.5	10	
D	25	64	

In all devices, the metal strip thickness is 10 nm Ti and 275 nm Au (identical deposition parameters), and the a-Si layer thickness is consistently 165 nm. This value is chosen to approximately match the penetration depth of a-Si at the 515 nm laser wavelength used in our experiments.

2. In the generation mechanism of terahertz waves, the author advocates that the main mechanism is the PGE (photovoltaic effect), but does not provide sufficient experimental evidence to rule out other possible mechanisms, such as the photon drag effect, the photothermal effect, the photo Dember effect, or the optical rectification effect. Although a preliminary analysis of these mechanisms is presented in the third paragraph on page 6 of the paper, further experimental validation is required. Especially when ruling out the photothermal effect, the judgment criteria of high-temperature conditions and time scale are debatable. First of all, the laser pumping itself can cause the thermal effect of the sample, leading to an increase in the

lattice temperature. Secondly, existing literature has pointed out that the WTe_2 material also exhibits the photothermal effect at low temperatures, and it is just more significant at high temperatures [Nat. Commun 2022, 13 (1), 3909]. Therefore, its occurrence at low temperatures should not be ignored. In addition, the difference in the time scale may not stem from the characteristics of the mechanism itself, but rather be limited by the sensitivity and time resolution of the detection system [Nat. Phys 2023, 19 (4), 507-514 cited by the author]. Therefore, the conclusion of ruling out the photothermal effect based solely on the existing analysis is not sufficient.

#Response#:

Currently, several experiments discuss the mechanism of directional photocurrent under bias-free conditions in WTe_2 and other similar semimetals. While early papers utilizing DC readout methods such as [1, 2, 3, 4, 5] argued that photocurrents originated from shift or photogalvanic currents, a number of recent papers argue that these signatures could instead occur due to an anisotropic photothermal effect (APTE) [6, 7]. Under our experimental conditions (20 K, picosecond time resolution), it is difficult to determine whether APTE is the only dominant component, without any contribution from the photogalvanic current. We cannot provide definitive evidence that resolves this ongoing discussion of the microscopic origin of the photocurrents at cryogenic temperatures and in highly non-equilibrium conditions. Therefore, in the text we will refer to it as *directional photocurrent*, which triggers cavity-related phenomena.

In more detail: As reported in Ref. [3], the so-called “robust edge current” in WTe_2 was observed and a possible mechanism of photogalvanic current was proposed based on symmetry arguments. Later, Ref. [6] performed a more detailed examination using nitrogen–vacancy centres and identified APTE as the underlying mechanism.

Both of these previous experiments were conducted at room temperature and in the DC-detection limit. When considering our low-temperature (20 K), ultrafast (picosecond time-resolution) measurements, two important questions arise regarding the microscopic mechanism: (i) How does the temperature dependence of the Seebeck coefficient influence APTE under our conditions? (ii) What is the timescale of APTE, and is it possible for the photogalvanic current to contribute on fast (picosecond) timescales?

As discussed in Ref. [6], the APTE is not determined by the absolute value of the Seebeck coefficient along the a - or b -axis (S_a , S_b), but rather by the difference between these two values. APTE becomes large only when $|S_a - S_b|$ is large.

To investigate APTE at low temperature, data are required for the temperature dependence of both S_a and S_b . However, due to limitations of the detection methods, a temperature dependence is usually measured along a single crystal axis. For example, the experimental work in Ref. [8] shows a clear temperature dependence of the Seebeck-coefficient along a -axis, but does not provide data for the difference between the a - and b -axes. In Ref. [6], a calculated temperature dependence of $|S_a - S_b|$ was provided, showing that $|S_a - S_b|$ tends to decrease to zero near 200 K and partially recovers towards 100 K, though no data were shown for temperatures below 100 K. We cannot comment on whether APTE is expected to dominate or is negligible in comparison to shift current generation below 100 K, but we are primarily interested in the fact that there is a directional photocurrent.

From the perspective of timescales, while we employ a time-resolved technique, we cannot directly correlate APTE to the picosecond-scale dynamics. Being a photothermal effect, the APTE timescale should be linked to the thermal decay time of WTe_2 , which has been measured to be on the order of a few picoseconds to several tens of picoseconds [9]. This leaves open the possibility that APTE could correspond to longer-lived components at the tail end of our measured signal, and thus contribute to the low-frequency emission.

Meanwhile, a possible photogalvanic current — if allowed by the symmetry breaking at the edges — would be expected to appear on picosecond timescales, which are within the detection capability of our setup.

More recently, Ref. [7] demonstrated the existence and non-equilibrium behaviour of the photothermal effect at various temperatures using similar techniques. The work provides supporting simulation data indicating that APTE can exist at both room temperature and low temperatures, serving as evidence that APTE can be present and pronounced under low-temperature and non-equilibrium conditions.

Yet, both our experiment and that of Ref. [7] face similar challenges in distinguishing photogalvanic current on the picosecond timescale. The available time resolution is limited by the response function of the photoconductive switch and the RC constant of the detection circuit, which are in the range of 1–2 ps. Our experimental setup is therefore not the ideal platform to conclusively determine the microscopic mechanism underlying the photocurrent generation.

The referee also mentioned other possible mechanisms, which we can more conclusively eliminate. The photon-drag effect typically occurs when photons transfer finite momenta to the electrons during photoexcitation, causing a DC current to flow. Because we photoexcite at normal incidence, any nonzero momentum transfer occurs in the z -direction of the material, which is orthogonal to our in-plane, xy detection scheme. We thus can rule out photon-drag effects based on geometric considerations [10, 11]. Similarly, the photo-Dember effect, which occurs due to the diffusion of carriers with unequal carrier mobilities, is also improbable in our geometry. This mechanism can result in in-plane emission at metal–material interfaces and would vanish at edges far from the metal strips, as well as reverse sign when the excitation position is shifted from near one stripline to near the opposite stripline [12, 3]. Photo-Dember effects also have been observed when there is a z -direction gradient of photoexcited carriers, however this will result in emission from carriers propagating normal to the surface of the material. As we are only sensitive to fields with in-plane fields, our sensing geometry precludes us from detecting z -direction currents [10]. The optical rectification current, which corresponds to the generation of a quasi-DC polarization in a nonlinear medium, is typically more significant in semiconductors or insulators, particularly when the excitation energy is below the material’s bandgap [10]. In our experiment, we expect the response is dominated by interband transitions, and compare the efficiency between the emission observed here and optical rectification in typical THz emitter crystals in the Supplementary Materials S.13.

3. In the position-dependent THz emission, is it sufficient to provide only the signals at the edge positions? Are there signals at other positions? Or is the signal minimum at the center and gradually increases as the position moves towards both sides? To better understand the radiation mechanism, it is recommended to supplement relevant data or diagrams in the attachment.

#Response#:

We observe that the photocurrent signal is maximized when the excitation spot is positioned on an edge. In the course of taking measurements shown in the paper, we frequently worked to maximize the detected signal. To do so, we monitored the lock-in output at a single time point corresponding to the peak of the photocurrent while translating the excitation beam position across the sample. It was repeatedly found that this photocurrent peak reached its highest value when the excitation was applied at the sample edge and progressively decreased as the excitation position was moved away from the edge.

To address the reviewer’s question and specifically examine the distinction between photocurrent contributions originating from the edges versus the bulk, we performed additional position- and fluence-dependent measurements on Device A. This new dataset has been included in the Supplementary Material (Section S.4).

The measurements include a third excitation position located in the central (non-edge) region between the metal strips. The upper and lower panels in the figure below correspond to the same excitation positions as shown in the main-text Fig. 2, while the middle panel presents the new data at the central area.

Figure R1: **Comparison of photocurrent response for edge and central excitation in Device A.** Upper and lower panels (a, c): excitation positions as in Fig. 2 of the main text; middle panel (c): measurement at the central (non-edge) region between the metal strips.

When excitation occurs between the metal strips, any possible cavity-related response is strongly damped (See Fig. R6 and related response section), leaving only the DC photocurrent component detectable. When exciting both between the metal strips and laterally in the middle of the flake, shown in the middle panel of Fig. R4, the photocurrent signal under similar excitation fluence is significantly suppressed compared to the edge positions. Due to the limited spatial resolution of our setup (determined by the pump-beam size and the optomechanical positioning system) and possible effects arising from the presence of the striplines (such as the formation of edge-like boundaries at the metal-strip edges), a perfectly zero signal is not observed. Nevertheless, the photocurrent is clearly maximized when the excitation position approaches the edges of the WTe_2 flakes.

4. The manuscript does not provide a clear explanation for the observed sign reversal of the THz field in device B under high pump fluence, as depicted in Figure 3. Or to be more specifically and deeply, why should the Purcell enhancement (or equivalently the resonant emission) only happen and be observed for high pump fluences? In our understanding, the Purcell enhancement effect only modulate the photonic/plasmonic density of state so to alter the emission rate for those affected frequencies. And Purcell effect is intrinsically

a linear effect, which means spectral modes are independent and the same enhanced emission spectra should be observed, rather than the power dependent enhancement observed in this manuscript. Therefore, the sign reversal and the nonlinear behavior for high excitation fluences indicate that the terahertz radiation signal in this work could be the result of complex combined action of multiple physical mechanisms, and current explanations are insufficient. Based on the analytical cavity model that the authors proposed, is it possible to simulate the flipping trend of the polarity of the terahertz radiation field under the dependence of the pump fluence, so as to provide support for the experimental results?

#Response#:

Generally, across the dataset, there are nonlinearities that alter the signal components between low and high fluence. The focus of this paper is largely on the shift in spectral weight from the DC photocurrent to the cavity resonances, which we describe as a Purcell enhancement. The analytical model that we employ is linear, and so while it can describe very well these shifts, it does not yet account for the additional nonlinearities the referee explains. We will comment further on some possible origins of these effects below, but believe a full treatment of the nonlinearities is beyond the scope of this paper, and is a topic that our work has opened up for future investigation.

Despite the existence of nonlinearities, the sign reversal of the THz field observed in Device B occurs due to a distinct reason. We therefore address this question in more depth in two parts:

1. The possibility of a nonlinear term in the Purcell factor, making the effect more pronounced at high fluence.
2. The observation of a sign reversal in Device B.

Nonlinear term in the Purcell factor: The referee pointed out that our analytical model remains a *linear* and *static* model, in which we have $j_{\text{cavity}} \propto j_{\text{ph}}$.

However, as demonstrated in several of our fluence-dependent measurements, the experimental data show evidence for a *nonlinear* relationship between j_{cavity} and j_{ph} . In particular, we consistently observe a stronger cavity-current response when increasing the pump fluence, indicating an effective enhancement of the Purcell factor at high fluence.

Given the high-fluence excitation fields involved in our experiments, nonlinear behaviour is not surprising. The peak excitation electric-field strength can reach a few MV/cm [13]. Such intensities place the system in a non-perturbative regime, where the photocurrent exhibits a linear-in- E dependence in the Rabi region. This regime implies that nonlinear correction terms may be required for a complete description of the Purcell factor.

Nonetheless, our current simplified analytical model already explains most observed phenomena and accurately predicts the Purcell resonance frequency. Attempts to introduce additional nonlinear terms would greatly complicate the model and introduce a large number of fitting parameters, which cannot be robustly constrained with our present dataset.

We note that fluence-dependence of the Purcell factor is still a cutting-edge problem, with only limited research in the field. To our knowledge, there is one experimental report of a power-dependent Purcell effect [14]. Theoretical studies also indicate that the Purcell factor can interplay with nonlinear processes, potentially exhibiting a threshold behaviour where it becomes large enough to significantly affect the system [15].

Various possible sources of nonlinearity in our system include:

1. Intensity-dependent corrections due to dynamical localisation [16, 17, 18, 19], i.e. E^2 flattening of the dispersion of different bands. This affects the density of states, which in turn can influence the second-order processes responsible for generating the photocurrent.
2. Formation of Floquet bands [20, 21, 22, 23, 13, 24]. This is similar to the above, but involves strong hybridisation with other bands. As seen in our previous work, this not only modifies the Bloch wavefunctions (and hence the associated matrix elements), but can also open new channels for dissipation, effectively amounting to Floquet bath engineering.
3. Nonlinear lattice effects [25, 26, 27, 28]. In WTe_2 there are several low-frequency Raman-active phonons associated with the topological phase. Such modes can be strongly optically rectified at large intensities, with potentially significant consequences for the underlying electronic structure.
4. Nonlinear behaviour of low-energy plasmon dynamics [29, 30, 31]. This is especially relevant when the current is amplified by forming a standing wave under the stripline. Such conditions can trigger complex nonlinear classical dynamics, potentially leading to effects such as self-focusing and self-amplification.

At present, adding the above nonlinearities into the model is highly complex and may have multiple, potentially unbounded physical origins. Our simplified model captures most of the observed phenomena and predicts the resonance frequency accurately. We have revised the main text to explicitly emphasize the distinction between the linear model and nonlinear effects observed.

Sign reversal in Device B: We now address the observed sign reversal between low and high fluence in the *time-domain* data for Device B. The time-domain signal in all cases (Figs. 3a and 3d) can be understood as a superposition of components at different frequencies, as shown in Figs. 3b and 3e. There is a phase inversion between the DC photocurrent and the Purcell-resonance peak, most likely due to a phase delay during the formation of the resonance.

In the *frequency domain*, as can be seen in the real part of the Fourier transform, a key difference between Device A and Device B is that, at high fluence, the photocurrent component remains pronounced in Device A but disappears from our detection in Device B, leaving only the Purcell contribution dominant. This is mainly because the Purcell resonance peak of Device B lies at a much lower frequency (≈ 0.25 THz). The existing spectral weight near that frequency is largely transferred from the DC photocurrent component to the Purcell component. This does *not* mean that the DC component is actually absent in Device B — the DC peak lies closer to zero frequency, but we cannot detect components below our 50 GHz limit. Thus, in the accessible detection band, it appears as though only the Purcell resonance remains.

In short: For Device B at low fluence, the DC photocurrent component dominates. At high fluence, only the Purcell resonance component, with a phase difference, is visible in our measurement band. The absence of DC in our detected spectrum does not imply that it vanishes physically — it is expected to remain strong at lower frequencies below 50 GHz — but in our measurable range, the DC-related current density has been converted into the Purcell-enhanced component, making the Purcell part the only feature we detect.

5. The Purcell factor is the strongest in device A in the manuscript, but it seems that the THz wave is not the strongest? What's the reason for this result? It is suggested that tables or graphs be provided in the attachment for comparison.

#Response#: The referee refers to Fig. S11 from the original manuscript, comparing Panel a with the remaining three panels. There are two points that require clarification:

1. Fig. S11, Panel a of the original manuscript showed a different type of data compared to the other panels. In the revised manuscript, we have separated these datasets into two figures, Fig. S13 and Fig. S14.
2. Although the current Fig. S13 illustrates a case comparable to the experiment, the quantitative experimental determination of the Purcell factor is not well determined and requires further study.

To address both points, we first restate a key detail: When fitting the Purcell resonance frequency in Fig. 4b, we do so *only* under the ideal limit, meaning that we assume the system damping is close to zero and the cavity has a very high quality factor. These assumptions do not hold in the actual experimental conditions.

From the experimental data, we observe a consistent redshift of the resonance peak with increasing fluence (see Figs. 3c and 3f). We fit this behaviour with a damped-harmonic-oscillator model, which yields the *undamped* resonance frequency (shown as dashed lines in those panels). This undamped frequency corresponds to the ideal case of the plasmonic cavity. To compare with theory, we use this undamped frequency rather than the peak frequency observed at a given fluence.

In the analytical model, the damping is represented by a parameter γ . In the updated version of the previous Fig. S11, we show simulated Purcell factors for all four devices using γ_{cav} as a damping conditions similar to those measured experimentally.

Figure R2: **Purcell factors for all devices using γ_{cavity} .** Panel **a** corresponds to Device A (same as Fig. 4a), Panel **b** shows Device B with a Purcell resonance near 0.2 THz, and similar results are shown for Devices C and D.

To illustrate the idealised situation, we also calculated the Purcell factor in the limit $\gamma \rightarrow 0$, producing a well-defined resonance frequency:

Figure R3: **Purcell factors in the ideal ($\gamma \rightarrow 0$) case for all devices.** These plots illustrate the Purcell factor for a highly idealized cavity in the limit of a very high quality factor, where the damping approaches zero.

The $\gamma \rightarrow 0$ case assumes a cavity with a very high quality factor corresponding to zero damping. Increasing γ (poorer quality factor) in the model also produces a redshift of the resonance peak; however, the shift predicted by this simplified model is much smaller than what is observed experimentally, which could be attributed to the nonlinear characteristics not captured by the theory.

Therefore, for a meaningful comparison between experiment and theory, Fig. 4b uses only the ideal-case limit: the undamped resonance frequencies extracted experimentally (dashed lines in Figs. 3c and 3f, lower panels) are compared with the resonance frequencies calculated in the $\gamma \rightarrow 0$ case of the analytical model.

Finally, in the theory the difference in resonance frequencies between the damped and undamped cases is relatively small; thus, Fig. 4a remains a good representation of both the width and the resonance frequency of the Purcell mode. However, although using γ_{cav} produces a case comparable to the experiment, this is not an quantitative determination of the *absolute* Purcell factor. The main reason is that we cannot simultaneously measure the amplitude of the photocurrent without cavity enhancement under identical conditions. We can estimate the photocurrent lineshape by using the case of excitation in between the striplines, but due to differences in detection sensitivity there exists an amplitude mismatch, which introduces uncertainty in calculating the value of $F = \left| \frac{j_{\text{cav}}}{j_{\text{ph}}} \right|$. We can state that the *magnitude* of the Purcell factor is similar in all cases, but determining its absolute value is beyond the scope of this work.

6. The Purcell-enhanced emission is not observed when the heterostructure is photoexcited between the

stripline traces. This observation indicates that the resonance standing-wave model proposed by the authors may be incomplete to describe the whole experimental results. In the model, the authors “assume that the local photocurrent excitation rapidly propagates throughout the device”, so we would expect that if the authors take the coefficients in the simulation of the case of “photoexcited outside the striplines” to simulate the case of “photoexcited inside the striplines”, the authors should also get some Purcell-enhancement result similar to the Purcell factor and the resonance frequency shown in Fig. 4. The fact that Purcell enhanced emission is not observed in this case, indicates that the model is incomplete or flawed. By the way, we noticed that in the equation of (23), there are three terms to sum to describe the cavity current j_{cav} : the right-going (reflected) current, the left-going (transmitted) current, and the intrinsic photocurrent itself. Whether the intrinsic photocurrent should still be counted here for the cavity current? Is there a double-count problem? The authors should clarify that.

#Response#: The referee raised two important questions regarding our analytical model:

1. Whether the model can explain the case of excitation inside the stripline where no Purcell enhancement is observed.
2. Whether Eq. (23) is written correctly.

1. Excitation inside the stripline and lack of Purcell enhancement The referee notes that, in principle, stronger enhancement should occur when the optical excitation spatially overlaps more closely with the cavity structure — i.e. excitation between the metal strips. This was indeed our initial approach, as shown in Fig. 2 and Fig. S7. However, in this configuration we did not observe an extra resonance peak; instead, only the DC photocurrent component was detectable.

We attribute this outcome to local heating induced by the pump laser. In our original manuscript, we proposed a qualitative explanation: direct excitation with the pump laser can cause local heating in the illuminated area, which increases the damping rate and drives the cavity into an overdamped regime when excited in the central region between strips. This heating effect is much less influential when the excitation is applied outside the stripline.

Our original analytical model in Section S.7 could describe excitation outside the stripline but did not fully explain why the Purcell resonance peak appears only in this geometry, not when exciting between metal strips.

To investigate this effect in greater detail, we extended our analytical model to include region-dependent damping coefficients, denoted γ_i .

Figure R4: **Cross section of the heterostructure with labeled regions.** The model assigns separate damping coefficients γ_i to each spatial region.

The geometry is divided into different regions (region 0 outside the stripline, region 2 inside the stripline gap). For areas not directly illuminated by the laser, we set $\gamma_i \equiv \gamma_{\text{cav}}$, corresponding to the experimentally extracted cavity damping. For areas directly subject to laser illumination, we compared two cases: 1. $\gamma_i = \gamma_{\text{cav}}$ (no additional heating effect), and 2. $\gamma_i = 50 \gamma_{\text{cav}}$ (representing a strong heating-induced increase in damping).

To estimate the possible effect of laser heating on the damping, we note that the optical constants of WTe₂ along the b -axis at 515 nm ($n \approx 3.2$, $k \approx 1.6$ [32]) give a reflectivity $R = \frac{(n-1)^2+k^2}{(n+1)^2+k^2} \approx 0.366$, corresponding to a worst-case absorbed fraction of $A_{\text{abs}} \approx 0.634$. The optical penetration depth is only $\delta \approx 5$ nm [33], so the absorbed energy density for a fluence F is $E_{\text{vol}} \approx FA_{\text{abs}}/\delta$. At low temperatures, the volumetric heat capacity follows the Debye law $C_{\text{vol}}(T) = \beta T^3$ with $\beta \approx 2.4 \times 10^{-5}$ J/(cm³ K⁴) [34, 35], giving the peak temperature after a single pulse as $T \approx \left(T_0^4 + \frac{4FA_{\text{abs}}}{\beta\delta}\right)^{1/4}$. For typical conditions ($T_0 = 20$ K, $F = 3$ mJ/cm²) this yields $T/T_0 \approx 8$.

In this material, the resistivity, ρ , scales as $\rho \propto T^2$ at low T [36]. The Drude model assumes that $\gamma \propto \rho$. Thus, one could expect that the linewidth will change as $\gamma/\gamma_0 \approx (T/T_0)^2 \approx 64$. This instantaneous heating thus could lead to an increase in damping on the order of 50, and thus we use in these calculations $\gamma \sim 50\gamma_0$ in the below estimates.

The calculated Purcell factors for Device A in all four cases are shown below:

Figure R5: **Simulated Purcell factors for Device A under various damping conditions.** Cases include excitation at region 0 (outside stripline) and region 2 (inside stripline gap), each evaluated with and without increased damping from local heating. The lower cut-off of the x-axis is 50 GHz, corresponding to the lower limit of our experimental bandwidth.

The simulation results show that when the laser excites region 0, the Purcell factor remains essentially unchanged in both low- and high-damping cases. This indicates that our detection is not strongly sensitive to extra damping outside the stripline gap.

In contrast, when the laser excites region 2 (inside the gap), the ideal case without additional damping still produces a Purcell enhancement of similar magnitude to the outside-excitation case. However, when heating increases the damping in region 2 ($\gamma_i = 50\gamma_{cav}$), the Purcell factor is strongly suppressed. Even with a more moderate damping enhancement of $\gamma_i = 8\gamma_{cav}$, the maximum Purcell factor remains only 1.1 when excitation occurs at the center, making it virtually undetectable in the raw data. This supports our previous qualitative picture: laser-induced heating in the gap lowers the cavity quality factor enough to push the system into an overdamped regime, eliminating the Purcell resonance peak. In this geometry, only the DC photocurrent component remains detectable, consistent with our experimental observations.

To better illustrate this mechanism, we provide a schematic timing diagram of the photocurrent generation and cavity interaction:

Figure R6: **Timing diagram of photocurrent generation.** (a) Excitation between metal stripes: local heating (yellow region) significantly increases damping and pushes the cavity into the overdamped regime, scattering the photocurrent during propagation and suppressing Purcell enhancement (as in Fig. 2 and Fig. S7). (b) Excitation outside the stripline: local heating still occurs in the illuminated area but mainly in an off-centre region; a substantial fraction of the photocurrent propagates into the screened region under the metal strip, allowing reflection across the cavity with minimal quality-factor reduction. As a result, Purcell enhancement is preserved (consistent with Fig. 3).

This extended modeling quantitatively confirms our qualitative reasoning: the suppression of Purcell enhancement for excitation between strips is due to local heating-induced overdamping, whereas excitation outside the stripline maintains a sufficiently high cavity quality factor for the enhancement to be observed.

2. Correctness of Eq. (23) Eq. (23) reflects a basic assumption of our simplified model, that it is both *linear* and *static*. The static aspect means that we assume an initial, time-invariant driving j_i . In the physical system, this corresponds to the photocurrent excited by the laser pulse. In response to this drive, the system exhibits reflection and transmission of the electromagnetic field, which we obtain by solving the boundary conditions. Because the solution is static, the driving term j_i does not decay, and since the model is linear, the final current expression is proportional to the initial drive j_i . If this term were omitted from the model, the solution would trivially reduce to zero for all currents.

From another perspective, we first assume that the initial driving term contains only a single frequency component ω . We then solve for the resulting current component at that same frequency. By varying ω , we obtain the full frequency spectrum of the cavity component.

Considering the above points, we believe that this manuscript is not suitable for publication in Nature Communications before being revised.

#Response#: We thank the referee once again for the helpful points and suggestions. In response, we have made corrections and modifications to the manuscript, and have added additional data where appropriate. All changes are marked with text in red in the revised version. We will be happy to address any further comments or questions, and we look forward to the referee's continued feedback.

References

- [1] Ma, J. *et al.* Nonlinear photoresponse of type-ii weyl semimetals. *Nature materials* **18**, 476–481 (2019).
- [2] Lim, S., Rajamathi, C. R., Süß, V., Felser, C. & Kapitulnik, A. Temperature-induced inversion of the spin-photogalvanic effect in WTe₂ and MoTe₂. *Physical Review B* **98**, 121301 (2018).
- [3] Wang, Q. *et al.* Robust edge photocurrent response on layered type ii weyl semimetal WTe₂. *Nature communications* **10**, 5736 (2019).
- [4] Zhang, Y. *et al.* Robust edge photogalvanic effect in thin-film WTe₂. *Applied Physics Letters* **125** (2024).
- [5] Mathew, A., Pulikodan, V. K. & Namboothiry, M. A. Understanding bulk photovoltaic effect in type-ii weyl semimetal Td-WTe₂ using polarization dependent photocurrent measurement. *Applied Physics Letters* **121** (2022).
- [6] Wang, Y.-X. *et al.* Visualization of bulk and edge photocurrent flow in anisotropic weyl semimetals. *Nature Physics* **19**, 507–514 (2023).
- [7] Chatterjee, S., Yoshioka, K., Wakamura, T., Perebeinos, V. & Kumada, N. Intrinsic ultrafast edge photocurrent dynamics in WTe₂ driven by broken crystal symmetry. *arXiv preprint arXiv:2510.06618* (2025).
- [8] Pan, Y. *et al.* Ultrahigh transverse thermoelectric power factor in flexible weyl semimetal WTe₂. *Nature communications* **13**, 3909 (2022).
- [9] Verma, S. *et al.* A room-temperature ultrafast carrier dynamical study and thickness-dependent investigation of WTe₂ thin films on a flexible pet substrate. *Physica Scripta* **99**, 105985 (2024).
- [10] Pettine, J. *et al.* Ultrafast terahertz emission from emerging symmetry-broken materials. *Light: Science & Applications* **12**, 133 (2023).
- [11] Maysonnave, J. *et al.* Terahertz generation by dynamical photon drag effect in graphene excited by femtosecond optical pulses. *Nano letters* **14**, 5797–5802 (2014).
- [12] Liu, C.-H. *et al.* Ultrafast lateral photo-dember effect in graphene induced by nonequilibrium hot carrier dynamics. *Nano letters* **15**, 4234–4239 (2015).
- [13] Li, X. *et al.* On-chip terahertz emission from floquet-bloch states. *Optical Materials Express* **15**, 1765–1776 (2025).
- [14] Canet-Ferrer, J. *et al.* Excitation power dependence of the purcell effect in photonic crystal microcavity lasers with quantum wires. *Applied Physics Letters* **102** (2013).
- [15] Tokman, M. *et al.* Purcell enhancement of the parametric down-conversion in two-dimensional nonlinear materials. *APL Photonics* **4** (2019).
- [16] Holthaus, M. Collapse of minibands in far-infrared irradiated superlattices. *Physical review letters* **69**, 351 (1992).
- [17] Lee, M. *et al.* Anderson localizations and photonic band-tail states observed in compositionally disordered platform. *Science Advances* **4**, e1602796 (2018).

- [18] Hübener, H., Sentef, M. A., De Giovannini, U., Kemper, A. F. & Rubio, A. Creating stable floquet–weyl semimetals by laser-driving of 3d dirac materials. *Nature communications* **8**, 13940 (2017).
- [19] Wang, Y., Steinberg, H., Jarillo-Herrero, P. & Gedik, N. Observation of floquet-bloch states on the surface of a topological insulator. *Science* **342**, 453–457 (2013).
- [20] Oka, T. & Kitamura, S. Floquet engineering of quantum materials. *Annual Review of Condensed Matter Physics* **10**, 387–408 (2019).
- [21] Zhou, L. *et al.* Cavity floquet engineering. *Nature communications* **15**, 7782 (2024).
- [22] Morimoto, T. & Nagaosa, N. Topological nature of nonlinear optical effects in solids. *Science advances* **2**, e1501524 (2016).
- [23] Matsyshyn, O., Piazza, F., Moessner, R. & Sodemann, I. Rabi regime of current rectification in solids. *Physical Review Letters* **127**, 126604 (2021).
- [24] Day, M. W. *et al.* Nonperturbative nonlinear transport in a floquet-weyl semimetal. *arXiv preprint arXiv:2409.04531* (2024).
- [25] Först, M. *et al.* Nonlinear phononics as an ultrafast route to lattice control. *Nature Physics* **7**, 854–856 (2011).
- [26] Sie, E. J. *et al.* An ultrafast symmetry switch in a weyl semimetal. *Nature* **565**, 61–66 (2019).
- [27] Hein, P. *et al.* Mode-resolved reciprocal space mapping of electron-phonon interaction in the weyl semimetal candidate Td-WTe₂. *Nature communications* **11**, 2613 (2020).
- [28] Ji, S., Granas, O. & Weissenrieder, J. Manipulation of stacking order in Td-WTe₂ by ultrafast optical excitation. *ACS nano* **15**, 8826–8835 (2021).
- [29] Stockman, M. I. Nanoplasmonics: past, present, and glimpse into future. *Optics express* **19**, 22029–22106 (2011).
- [30] Maier, S. A. *et al.* *Plasmonics: fundamentals and applications*, vol. 1 (Springer, 2007).
- [31] Kauranen, M. & Zayats, A. V. Nonlinear plasmonics. *Nature photonics* **6**, 737–748 (2012).
- [32] Munkhbat, B., Wróbel, P., Antosiewicz, T. J. & Shegai, T. O. Optical constants of several multilayer transition metal dichalcogenides measured by spectroscopic ellipsometry in the 300–1700 nm range: high index, anisotropy, and hyperbolicity. *ACS photonics* **9**, 2398–2407 (2022).
- [33] Buchkov, K. *et al.* Anisotropic optical response of WTe₂ single crystals studied by ellipsometric analysis. *Nanomaterials* **11**, 2262 (2021).
- [34] Callanan, J. E., Hope, G., Weir, R. D. & Westrum Jr, E. F. Thermodynamic properties of tungsten ditelluride (WTe₂) i. the preparation and low temperature heat capacity at temperatures from 6 k to 326 k. *The Journal of Chemical Thermodynamics* **24**, 627–638 (1992).
- [35] Laboratory, Q. M. O. Tungsten ditelluride (wte₂) optical properties database (2024). URL <https://quantumlab.uark.edu/wte2/>. Accessed: 2024-07-06.
- [36] Perevalova, A. *et al.* Electronic transport in a topological semimetal WTe₂ single crystal. *arXiv preprint arXiv:2302.00297* (2023).

Reviewer 4 & 5

The manuscript investigates photocurrent generation in a layered 2D van der Waals self-cavity. A photocurrent is observed via ultrafast optoelectronic circuitry when light is shone at the edge of the sample where mirror symmetry is broken. An enhancement of the photocurrent is reported when light is shone outside the cavity, which the authors attribute to Purcell effects.

From a technological standpoint, the authors employ ultrafast optoelectronic circuitry that avoids issues associated with voltage biases and can reliably detect electric signals at terahertz frequencies. This is quite an instrumental and experimental feat: a technique first pioneered by some of the authors (e.g., in Ref. 18). The authors have now extended this technique to WTe₂. Overall, the manuscript is well-written, supported by detailed analysis, and presents promising applications in next-generation THz technologies that could appeal to a broad audience. I really like it. However, we do have some comments that we hope the authors elaborate on:

#Response#: We thank the referee for the time and effort spent reviewing our manuscript and for their thoughtful, constructive comments. These have been valuable in improving the clarity of the work and presenting the novelty and impact more effectively. In response to the referee’s questions and suggestions, we have made significant changes to both the main text and the supplementary information. All changes are highlighted in red in the revised manuscript for clarity. Below we address each comment point-by-point.

1. One major question I have is: to what extent is this truly an enhancement? This ties directly to the authors’ claim that the photocurrent excited inside the cavity does not experience such enhancement (why is that?). For instance, if I compare the red line in Fig. 2b (or S5) with Fig. 3a, at a similar fluence of 1.64 and 1.77 mJ/cm² respectively, the only difference is that, in the former case, light shines between the striplines, while in the latter, it shines outside them. Yet, the THz emission peak in the former case is about 3-4 times larger than in the latter. If, as the authors claimed, this is an enhancement, shouldn’t I expect the former to be smaller? Or could it be that when light shines inside the striplines, the enhancement is still there, but the frequency of the peak of Purcell factors is somewhat suppressed? And if so, why is that? A clear and simple explanation of what the simple experimental “smoking gun” of the Purcell enhancement is would be very nice, especially given this seeming difference between no Purcell enhancement inside the striplines, but only there outside the striplines. While I understand that a detailed explanation might be outside the scope of this current work, it leaves a rather unsettling taste that it is not clear what the signature is.

#Response#:

The enhancement we define here is a frequency-dependent enhancement — specifically, the appearance of a new resonance peak due to the plasmonic cavity. If the sample were a pure flake without such a cavity, which is comprised of both the flake and the metal transmission lines, the frequency-domain response would be essentially trivial, showing only the photocurrent peak. Using our analytical model, we can estimate the expected frequency of this resonance peak, which also allows us to design the flakes to set and control the resonance frequency.

For Devices A and B, we carefully selected the thicknesses of the hBN and WTe₂ layers to place the resonance frequency near 0.55 THz and 0.2 THz, respectively. The agreement between the experimentally measured Purcell resonance frequencies and the values predicted by the analytical model (main-text Fig. 4b) serves as the “smoking gun” evidence for the Purcell enhancement.

To answer this question in more detail, we separate the discussion into three parts:

1. How we define the enhancement.
2. The difference between excitation inside versus outside the striplines.
3. The absolute amplitude difference between these two excitation geometries.

1. Our definition of enhancement

The enhancement we refer to is not a comparison between different measurements, but rather an increase in the photocurrent density at a specific frequency resulting from the presence of the plasmonic cavity structure.

Specifically, comparing the frequency components of the DC photocurrent in Fig. S5 with those after Purcell enhancement in Fig. 2b reveals the emergence of a *new resonance peak* (Fig. 2) — corresponding to an increased current density near the plasmonic resonance — absent in the pure DC photocurrent case. We refer to this frequency-specific increase in current density (and thus THz emission) as **Purcell enhancement**.

In our devices, the presence of the metal strips enables a **shift in frequency** of the current density, which we describe via Fermi’s golden rule (Eq. 2 of the main text):

$$\Gamma_{j_{\text{ph}} \rightarrow j_{\text{cav}}} \propto |\langle j_{\text{cav}}(\omega) | H' | j_{\text{ph}}(\omega) \rangle|^2 \rho(\omega),$$

As shown in the Purcell factor simulations in Fig. 4a, the resonance corresponds to the plasmon frequency of the plasmonic cavity. A Purcell factor greater than unity means that the current density at the corresponding frequency is larger than it would be without the stripline present.

Physically, this is not simply an amplification of the DC component, but rather a *re-distribution* of current density from frequencies where the Purcell factor is less than one (in our devices mainly near DC) toward the resonance frequency.

2. Difference between inside and outside excitation

The referee notes that, in principle, stronger enhancement should occur when the optical excitation spatially overlaps more closely with the cavity structure — i.e. excitation between the metal strips. This was indeed our initial approach, as shown in Fig. 2 and Fig. S7. However, in this configuration we did not observe an extra resonance peak; instead, only the DC photocurrent component was detectable.

We attribute this outcome to local heating induced by the pump laser. We previously proposed a qualitative explanation: direct excitation with the pump laser can cause local heating in the illuminated area, which increases the damping rate and drives the cavity into an overdamped regime when excited in the central region between strips. This heating effect is much less influential when the excitation is applied outside the stripline.

Our original analytical model in Section S.7 could describe excitation outside the stripline but did not fully explain why the Purcell resonance peak appears only in this geometry, not when exciting between metal strips.

To investigate this effect in greater detail, we extended our analytical model to include region-dependent damping coefficients, denoted γ_i .

Figure R1: **Cross section of the heterostructure with labeled regions.** The model assigns separate damping coefficients γ_i to each spatial region.

The geometry is divided into different regions (region 0 outside the stripline, region 2 inside the stripline gap). For areas not directly illuminated by the laser, we set $\gamma_i \equiv \gamma_{\text{cav}}$, corresponding to the experimentally extracted cavity damping. For areas directly subject to laser illumination, we compared two cases: 1. $\gamma_i = \gamma_{\text{cav}}$ (no additional heating effect), and 2. $\gamma_i = 50 \gamma_{\text{cav}}$ (representing a strong heating-induced increase in damping).

To estimate the possible effect of laser heating on the damping, we note that the optical constants of WTe_2 along the b -axis at 515 nm ($n \approx 3.2$, $k \approx 1.6$ [1]) give a reflectivity $R = \frac{(n-1)^2 + k^2}{(n+1)^2 + k^2} \approx 0.366$, corresponding to a worst-case absorbed fraction of $A_{\text{abs}} \approx 0.634$. The optical penetration depth is only $\delta \approx 5$ nm [2], so the absorbed energy density for a fluence F is $E_{\text{vol}} \approx F A_{\text{abs}} / \delta$. At low temperatures, the volumetric heat capacity follows the Debye law $C_{\text{vol}}(T) = \beta T^3$ with $\beta \approx 2.4 \times 10^{-5} \text{ J}/(\text{cm}^3 \text{ K}^4)$ [3, 4], giving the peak temperature after a single pulse as $T \approx \left(T_0^4 + \frac{4FA_{\text{abs}}}{\beta\delta} \right)^{1/4}$. For typical conditions ($T_0 = 20$ K, $F = 3 \text{ mJ}/\text{cm}^2$) this yields $T/T_0 \approx 8$.

In this material, the resistivity, ρ , scales as $\rho \propto T^2$ at low T [5]. The Drude model assumes that $\gamma \propto \rho$. Thus, one could expect that the linewidth will change as $\gamma/\gamma_0 \approx (T/T_0)^2 \approx 64$. This instantaneous heating thus could lead to an increase in damping on the order of 50, and thus we use in these calculations $\gamma \sim 50\gamma_0$ in the below estimates.

The calculated Purcell factors for Device A in all four cases are shown below:

Figure R2: **Simulated Purcell factors for Device A under various damping conditions.** Cases include excitation at region 0 (outside stripline) and region 2 (inside stripline gap), each evaluated with and without increased damping from local heating. The lower cut-off of the x-axis is 50 GHz, corresponding to the lower limit of our experimental bandwidth.

The simulation results show that when the laser excites region 0, the Purcell factor remains essentially unchanged in both low- and high-damping cases. This indicates that our detection is not strongly sensitive to extra damping outside the stripline gap.

In contrast, when the laser excites region 2 (inside the gap), the ideal case without additional damping still produces a Purcell enhancement of similar magnitude to the outside-excitation case. However, when heating increases the damping in region 2 ($\gamma_i = 50\gamma_{cav}$), the Purcell factor is strongly suppressed. Even with a more moderate damping enhancement of $\gamma_i = 8\gamma_{cav}$, the maximum Purcell factor remains only 1.1 when excitation occurs at the center, making it virtually undetectable in the raw data. This supports our physical picture: laser-induced heating in the gap lowers the cavity quality factor enough to push the system into an overdamped regime, eliminating the Purcell resonance peak. In this geometry, only the DC photocurrent component remains detectable, consistent with our experimental observations.

To better illustrate this mechanism, we provide a schematic timing diagram of the photocurrent generation and cavity interaction:

Figure R3: **Timing diagram of photocurrent generation.** (a) Excitation between metal stripes: local heating (yellow region) significantly increases damping and pushes the cavity into the overdamped regime, scattering the photocurrent during propagation and suppressing Purcell enhancement (as in Fig. 2 and Fig. S7). (b) Excitation outside the stripline: local heating still occurs in the illuminated area but mainly in an off-centre region; a substantial fraction of the photocurrent propagates into the screened region under the metal strip, allowing reflection across the cavity with minimal quality-factor reduction. As a result, Purcell enhancement is preserved (consistent with Fig. 3).

This extended modeling quantitatively confirms our qualitative reasoning: the suppression of Purcell enhancement for excitation between strips is due to local heating-induced overdamping, whereas excitation outside the stripline maintains a sufficiently high cavity quality factor for the enhancement to be observed.

3. Absolute amplitude difference in DC photocurrent detection

There is also an absolute amplitude difference for the DC photocurrent component when measured in the inside versus outside waveguide. This is mainly a matter of signal-detection sensitivity. For the signal to be detected by the photoconductive switch, the emitted THz field must couple to the *odd mode* of the coplanar stripline, which has stronger field confinement between the metal strips and much weaker fields outside [6]. This means that, even if the material generates the same magnitude of photocurrent, photocurrent generated *between* the metal strips is coupled more efficiently into the detection system, resulting in a larger measured amplitude. However, as mentioned previously, excitation at the centre also damps out the cavity resonance due to heating.

Thus, when excitation is at the centre, we measure a signal of higher amplitude — because the system collects the photocurrent emission more efficiently — but the signal contains *only* the DC photocurrent component, with the Purcell resonance suppressed.

2. A little confusion in Fig. 3: device A exhibits a peak near 0.1 THz (attributed to DC photocurrent) that is larger in amplitude than the Purcell-enhanced peak at 0.4 THz, whereas this “DC peak” is absent in device B, making the latter appear more ideal. Could this difference be explained in terms of factors such as sample edge alignment, sample thickness, or the size of the light beam? In fact, how do the authors identify which peaks experience the Purcell enhancement: these seemed to be different features for device A and device B. To that point, what extent is the observed enhancement engineerable? Could the authors provide

some qualitative insights, based on experimental trends or theoretical considerations, on how the frequency and amplitude of the enhancement might be tuned?

#Response#: To answer this question clearly, we separate the discussion into three points:

1. How we distinguish the Purcell resonance peak from the DC photocurrent component.
2. Why Device B appears to omit the DC component.
3. How the Purcell resonance frequency can be tuned.

1. Distinguishing the Purcell resonance from DC components

We rely on two main observations:

First, the fluence dependence. In our experiments, at low fluence the photocurrent component is always dominant, whereas at high fluence an additional component becomes apparent. Comparing the frequency-domain data in Fig. 3b and Fig. 3e reveals that the photocurrent peak near DC (dominant at low fluence) and the Purcell resonance peak at higher frequency (more obvious at high fluence) consistently have opposite signs. This may be due to a phase delay associated with the formation of the cavity resonance, and serves as evidence for the existence of a new mode.

Secondly, the newly appeared dynamic matches the analytical model for all four devices. From the experimental data, we also observe a consistent redshift of the resonance peak with increasing fluence (Figs. 3c and 3f). We fit this behaviour using a damped-harmonic-oscillator model, which yields the *undamped* resonance frequency (shown as dashed lines in those panels). Using this undamped frequency as the ideal limit, we compare it with the Purcell resonance frequency calculated analytically for all four devices and find good agreement (Fig. 4b). This agreement supports the identification of the additional mode as the Purcell resonance.

2. Why Device B appears to omit the DC component

The observed sign reversal between low and high fluence in the *time-domain* data for Device B can be understood by considering the superposition of frequency components. The time-domain signals (Figs. 3a and 3d) are composed of contributions at different frequencies, as illustrated by the spectra in Figs. 3b and 3e.

A key distinction between Device A and Device B is that, at high fluence, the photocurrent component remains clearly present in Device A but disappears from our detectable range in Device B, leaving only the Purcell contribution visible. This is primarily because the Purcell resonance peak of Device B lies at a much lower frequency (≈ 0.25 THz) [7]. The spectral weight near that frequency is largely transferred from the DC photocurrent component to the Purcell component. This does *not* mean the DC component is physically absent in Device B — the DC peak lies closer to zero frequency, but components below our 50 GHz detection threshold cannot be measured. Thus, within our detection bandwidth, it appears that only the Purcell resonance remains.

In short: For Device B at low fluence, the DC photocurrent component dominates. At high fluence, only the Purcell resonance component — with a phase difference — is visible in our measurement band. The apparent absence of DC in the detected spectrum is due to our bandwidth limitation; the DC-related current density is shifted into the Purcell-enhanced frequency range, leaving only the Purcell part in the accessible band.

3. Tunability of the Purcell resonance frequency

To illustrate how the Purcell resonance frequency can be tuned, we present a simplified version of the analytical theory. A precise determination of the resonance frequency requires solving the full boundary

equations described in Section S.7. However, it can be interpreted qualitatively as a superposition of the resonance frequencies for the screened and unscreened plasmon modes, corresponding to regions with and without the metal strips, respectively. The screened plasmon frequency is the dominant term and can be expressed as:

$$\omega_{\text{screened}} = \sqrt{\frac{\omega_{3D}^2 q^2 d_{\text{WTe}_2} d_{\text{hBN}}}{\epsilon_{\text{hBN}}}}, \quad (1)$$

where ω_{3D} is the 3D plasmon frequency along the detection direction (perpendicular to the stripline), d denotes the thickness of each layer, and the wavevector $q \approx \frac{2\pi}{4W_1}$ is roughly determined by W_1 , the width of one metal strip.

While this simplified equation does not yield an exact prediction of the plasmon frequency, it gives a useful sense of how the resonance can be tuned. During fabrication, the most straightforward parameters to control are the thicknesses of the WTe₂ and hBN layers, and the crystal orientation of the WTe₂ flake. For example, in designing Device A, we selected these parameters to position the resonance frequency near the centre of our detection bandwidth (≈ 0.5 THz).

In response to the referee’s question, we have added Section S.7.1 to the revised manuscript, which provides a detailed description of this point.

3. The authors say that the signal is generated by a photogalvanic current (Sec. 2.2) - I take this to mean a current that is generated not from a p-n junction origin (the authors later suggest that it is from shift current in Sec. 3). Given the importance of the striplines in the experimental set-up, is there an easy way of understanding how the striplines do not contribute to p-n junction-like photocurrent? Saying this clearly will really help the untrained reader zero in on how to say something is of p-n junction origin and what isn’t.

#Response#: The photogalvanic current referred to here is not analogous to the current generated at the interface of a p - n junction, but rather a directional current that can exist when the crystal lacks inversion symmetry [8, 9]. In a crystal with broken inversion symmetry, such a directional current can be generated uniformly throughout the bulk. For example, WTe₂ can exhibit such behaviour along its c -axis.

For the in-plane case of WTe₂ (along the a - and b -axes), the situation is more specific due to the presence of two mirror planes in orthogonal directions, M_a and \tilde{M}_b . The coexistence of these two mirror symmetries prevents the formation of directional current within the bulk (in-plane direction) by symmetry constraints. Only at edges, where one or both of these symmetries are broken, is the photogalvanic current allowed by symmetry.

The referee is correct that the interface between the metal strip and the material could influence the photocurrent generation process. However, such interface-driven generation would be strongly position dependent and would typically lead to inversion of the current sign when the excitation beam is moved in the direction perpendicular to the waveguide. We do not observe such behaviour, and therefore conclude that this component is not dominant in our measurements.

A recent experiment by Chatterjee [7], using a similar technique, has shown that photocurrent generated at the metal–material interface occurs on a much longer timescale than our time resolution of \sim ps. This suggests that such interface-related contributions are not relevant to our data.

Nonetheless, we emphasise that our experimental setup is not ideally suited to conclusively determine the microscopic mechanism underlying the photocurrent generation — particularly in distinguishing between photogalvanic current and the anisotropic photothermoelectric effect [10, 7].

#Response#: We thank the referee once again for the helpful points and suggestions. In response, we have made corrections and modifications to the manuscript, and have added additional data where appropriate. All changes are marked with text in red in the revised version. We will be happy to address any further comments or questions, and we look forward to the referee’s continued feedback.

References

- [1] Munkhbat, B., Wróbel, P., Antosiewicz, T. J. & Shegai, T. O. Optical constants of several multilayer transition metal dichalcogenides measured by spectroscopic ellipsometry in the 300–1700 nm range: high index, anisotropy, and hyperbolicity. *ACS photonics* **9**, 2398–2407 (2022).
- [2] Buchkov, K. *et al.* Anisotropic optical response of WTe₂ single crystals studied by ellipsometric analysis. *Nanomaterials* **11**, 2262 (2021).
- [3] Callanan, J. E., Hope, G., Weir, R. D. & Westrum Jr, E. F. Thermodynamic properties of tungsten ditelluride (WTe₂) i. the preparation and low temperature heat capacity at temperatures from 6 k to 326 k. *The Journal of Chemical Thermodynamics* **24**, 627–638 (1992).
- [4] Laboratory, Q. M. O. Tungsten ditelluride (wte₂) optical properties database (2024). URL <https://quantumlab.uark.edu/wte2/>. Accessed: 2024-07-06.
- [5] Perevalova, A. *et al.* Electronic transport in a topological semimetal WTe₂ single crystal. *arXiv preprint arXiv:2302.00297* (2023).
- [6] Karnetzky, C. *et al.* Towards femtosecond on-chip electronics based on plasmonic hot electron nano-emitters. *Nature communications* **9**, 2471 (2018).
- [7] Chatterjee, S., Yoshioka, K., Wakamura, T., Perebeinos, V. & Kumada, N. Intrinsic ultrafast edge photocurrent dynamics in WTe₂ driven by broken crystal symmetry. *arXiv preprint arXiv:2510.06618* (2025).
- [8] Wang, Q. *et al.* Robust edge photocurrent response on layered type ii weyl semimetal WTe₂. *Nature communications* **10**, 5736 (2019).
- [9] Xie, X. *et al.* Surface photogalvanic effect in ag₂te. *Nature Communications* **15**, 5651 (2024).
- [10] Wang, Y.-X. *et al.* Visualization of bulk and edge photocurrent flow in anisotropic weyl semimetals. *Nature Physics* **19**, 507–514 (2023).

Authors Response to Reviews

March 1, 2026

Reviewer 1

The authors have performed additional measurements and provided clarifications addressing my questions, which I appreciate. While some aspects of the interpretation do not yet fully align with the current Purcell picture, the experimental observations are robust. I therefore support publication of the revised manuscript.

#Response#: We thank the referee for their careful reading of the revised manuscript and for their positive and constructive assessment. We are pleased that the referee considers the manuscript suitable for publication.

Reviewer 2 & 3

In the revised manuscript, some improvements have been made to address some of my questions. However, I am still hesitant to recommend its publication in Nature Communications. The detailed reasons are listed below:

#Response#: We sincerely thank the referee for their continued time and effort in reviewing our manuscript, and for the additional thoughtful and constructive questions raised in this second round. These further comments have been valuable in refining the presentation and ensuring the robustness of our conclusions. In response, we have made additional clarifications and adjustments in both the main text and the supplementary information. All new changes are highlighted in red in the revised manuscript for ease of reference. Below, we address each of the referee's points in detail, point-by-point.

(1)Based on the thickness of devices characterized by AFM, it is evident that there is a substantial difference in the thickness of WTe₂ among these devices. In particular, for device D, the thickness difference is nearly an order of magnitude. Wouldn't this difference in thickness have an impact on the experimental results? The thickness of WTe₂ is included in the model, which makes it highly questionable whether thickness affects the overall experimental results or not. Moreover, according to the Purcell factor simulation results of device D in Figs. S13 and S14 in the supplementary materials, it is evident that the simulation results of device D differ significantly from those of other devices.

#Response#:

The Referee is correct that the flake thickness has a notable impact on the experimental results. However, the primary impact of flake thickness is explicitly accounted for in the model's predictions of the self-cavity

plasma frequency, and is a key factor in the agreement between model and experiment for Devices A-C. The model of plasmonic cavity frequencies was experimentally validated through a further 22 graphene or graphite-based devices in ref [1]. In this work, we apply this model to the calculation of self-cavity THz emission. While there remains a discrepancy between model and experiment for Device D, we quantitatively demonstrate below that a large portion of this difference results trivially from the limited bandwidth of our measurement. The remainder of the discrepancy likely comes from the assumptions of the model (which approximates the sample thickness as a linear electrodynamic effect) beginning to break down at the higher thickness of the sample in Device D. In our view, Device D exhibits sufficient agreement with the model while also demonstrating what happens when pushing the limits of both model and experiment. If the Referee prefers, however, we would be happy to move all data and discussion of Device D to the supplement. We elaborate on each of these points in more detail below.

In our analytical and numerical models, the WTe₂ thickness is indeed an important factor influencing the plasmonic resonance frequency, along with other geometrical parameters such as the hBN thickness and the transmission line width. To illustrate the role of thickness, we provide a simplified derivation of the Purcell resonance frequency based on our analytical theory (full boundary-condition solution in Section S.8). The resonance can be qualitatively understood as a superposition of the frequencies of screened and unscreened plasmon modes, corresponding to regions with and without the metal strips, respectively. The dominant screened-plasmon frequency can be written as:

$$\omega_{\text{screened}} \approx \sqrt{\frac{\omega_{3\text{D}}^2 q^2 d_{\text{WTe}_2} d_{\text{hBN}}}{\epsilon_{\text{hBN}}}}, \quad (1)$$

where $\omega_{3\text{D}}$ is the 3D plasmon frequency along the detection direction (perpendicular to the stripline), d is the layer thickness, and $q \approx \frac{2\pi}{4W_1}$ is set primarily by W_1 , the width of one metal strip (Eq. S40 in Section S.8.1).

Equation 1 would correspond exactly to the Purcell resonance in the limit $W_0, W_2, W_4 \rightarrow 0$, where only the screened region remains. In this case,

$$\lim_{W_0, W_2, W_4 \rightarrow 0} \omega_{\text{Purcell}} = \omega_{\text{screened}}. \quad (2)$$

Under experimental conditions, however, the widths W_0 , W_2 , and W_4 are finite, and the Purcell resonance is therefore shifted due to hybridisation between screened and unscreened plasmon modes within the cavity. Nevertheless, ω_{screened} still provides a useful reference for estimating the overall trend of the Purcell resonance frequency:

$$\omega_{\text{Purcell}} \sim \omega_{\text{screened}}. \quad (3)$$

Although this expression is simplified and not quantitatively exact, it captures the main tunability of the resonance through controllable fabrication parameters such as the thicknesses of WTe₂ and hBN and the crystal orientation of the WTe₂ flake. For example, in Device A we selected these parameters to center the resonance at ≈ 0.5 THz, within our detection bandwidth.

The mismatch between the model and the experimental data shown in Fig. 4b can be partly attributed to the frequency-dependent response function of the photoconductive switches used for detection. In an ideal scenario, a perfectly flat detection response in the frequency domain would allow the measured signal to directly reflect the intrinsic emission spectrum of the plasmonic mode. In practice, however, the response function of the photoconductive switches is non-uniform, and decreases at higher frequencies.

The red curve in Fig. 1 shows a reference response trace of the photoconductive switch, obtained by

triggering THz emission from one photoconductive switch, propagating the signal through an empty circuit, and detecting it with a second switch. This reference measurement reveals a maximum detection sensitivity around 0.2 THz, followed by a continuous decay at higher frequencies, dropping below $\sim 20\%$ at ~ 0.8 THz. This behaviour effectively sets the upper limit of our detection bandwidth.

Such a non-uniform response function modifies the measured spectra and can shift the apparent resonance peak frequency. The magnitude of this shift depends significantly on the intrinsic linewidth (FWHM) of the resonance. To illustrate this effect, Fig. 1 shows a simple simulation in which the intrinsic plasmonic resonance is modelled as a Gaussian peak, $\exp\left[-\frac{(\omega-\omega_0)^2}{2\sigma^2}\right]$, where ω_0 is the calculated Purcell resonance frequency (as used in Fig. 4b) and $\sigma = \text{FWHM}/(2\sqrt{2\ln 2}) \approx \text{FWHM}/2.35$, with the FWHM extracted from experimental fits. Multiplication of this Gaussian spectrum by the measured switch response function results in a distorted peak with a shifted maximum (blue curves in Fig. 1).

Figure 1: **Influence of the frequency-dependent detector response on the apparent plasmonic resonance peak.** Panels (a)–(d) correspond to Devices A–D, respectively. All traces are normalised for clarity. The red curves show a reference frequency response of the photoconductive switch, obtained by generating a THz pulse from one photoconductive switch, propagating it through an empty circuit, and detecting it with a second switch. The black curves represent the intrinsic plasmonic resonance modelled as a Gaussian peak centred at the calculated Purcell resonance frequency ω_0 , with the linewidth (FWHM) extracted from experimental fits. The blue curves show the resulting detected spectra obtained by multiplying the intrinsic resonance by the measured detector response function, i.e. $S(\omega) = G(\omega)R(\omega)$. The apparent shift of the peak position arises from the frequency dependence of the detector response and increases with both the intrinsic linewidth and the local slope of the response function at ω_0 , consistent with $\omega_{\text{peak}} - \omega_0 = \sigma^2(d \ln R/d\omega)|_{\omega_0}$.

The estimated peak shifts for all devices are summarised in Table 1. The detected spectrum can be written as $S(\omega) = G(\omega)R(\omega)$, where $G(\omega)$ is the intrinsic plasmonic resonance and $R(\omega)$ is the frequency-dependent response function of the photoconductive switch. The peak position is therefore determined by $\frac{d}{d\omega} \ln S(\omega) =$

$\frac{d}{d\omega} \ln G(\omega) + \frac{d}{d\omega} \ln R(\omega) = 0$. For a Gaussian resonance, $G(\omega) = \exp\left[-\frac{(\omega-\omega_0)^2}{2\sigma^2}\right]$, one obtains $\omega_{\text{peak}} - \omega_0 = \sigma^2 \left. \frac{d}{d\omega} \ln R(\omega) \right|_{\omega_0}$.

Within this model, Device D exhibits the largest apparent shift for two reasons. First, it has a comparatively large intrinsic linewidth (large σ), which amplifies the influence of the response function. Second, its resonance frequency ($\omega_0 \approx 1.15$ THz) lies in a spectral region where the response function $R(\omega)$ varies rapidly, i.e. where $d \ln R/d\omega$ is large. The combination of a large σ and a steep local slope of the response function therefore leads to the pronounced peak shift observed for Device D.

Table 1: Estimated resonance peak shifts induced by the non-uniform frequency response of the photoconductive switch.

Device	ω_0 (THz)	FWHM (THz)	Peak shift (THz)
A	0.54	0.45	0.09
B	0.20	0.28	~ 0
C	0.41	0.28	0.03
D	1.15	0.49	0.23

This correction is not included in Fig. 4b. The response function shown here is obtained from a reference measurement using two photoconductive switches and is expected to be representative but not identical to the effective response in all emission experiments presented in this work. Accurately determining the exact response function for each measurement configuration is not trivial. While the response function above adds intuition and further illustrates the effect of the detection bandwidth on the measured resonances, the correction is expected to be small in devices A-C and within other sources of experimental uncertainty. For this reason, we do not apply an explicit response-function correction in Fig. 4b, but note that the largest uncertainty is for Device D. Should the referee prefer, we can include this discussion in the supplement.

Also, it is indeed possible that increasing the WTe₂ thickness (by roughly half an order of magnitude) introduces additional variations beyond those captured by our simplified model. A primary consideration is the modification of the effective Coulomb screening. In the present theoretical treatment, increasing the thickness effectively amounts to stacking multiple layers while retaining an essentially two-dimensional interaction form, for which the Coulomb kernel scales as $\tanh(qd)/q \sim 1/q$ in the long-wavelength limit. However, once the sample thickness becomes sufficiently large that out-of-plane Coulomb interactions are no longer negligible, the screening is expected to cross over towards a three-dimensional form, scaling as $\sim 1/q^2$. Such enhanced screening would reduce the plasmon dispersion to lower frequencies compared to the predictions of a purely two-dimensional model [2, 3, 4, 5]. More generally, this behaviour is well established for plasmons in van der Waals materials, where dielectric screening and geometric confinement play central roles in determining plasmon dispersion and confinement [4, 5, 6]. A more quantitative assessment of these thickness-dependent effects would require future measurements with an extended detection bandwidth.

Device D was the first device that we investigated, but back-of-the-envelope calculations indicated that its emission peak was likely outside our detection range. Based on these initial calculations, we revised the design with optimised parameters and fabricated Devices A and B, in which the signal is much better resolved and the Purcell peak frequency agrees well with the model.

Nevertheless, Device D still follows the overall expected trend and helps account for the discrepancy between the theoretical prediction and the experimental result shown in Fig. 4b. We therefore include Device D for full transparency regarding the experimental development and validation process. Given that the mismatch for Device D is substantially larger due to bandwidth and detector-response limitations, we

have added this additional discussion into the Supplementary information.

(2) Although the author downplays the mechanism of terahertz emissions and attributes it to the directional edge photocurrent, the thermal effect has not been explicitly excluded. However, the theoretical derivation in the supplementary material still clearly uses the PGE effect for explanation. Moreover, the representation of this nonlinear susceptibility tensor appears to be incorrect, and the author should carry out a detailed examination.

#Response#: We agree that thermal effects cannot be excluded, and thus we changed the title of the paper and added further analysis and discussion in the first revision. However, we recognise that Section S.1 of the Supplementary Information was not explicitly updated at that stage and thank the referee for raising this point. We have now thoroughly revised Section S.1 in the updated manuscript to include an analysis of allowed thermal and photogalvanic currents based on crystal symmetries, and present the updated text in full below.

In this revised version, we have renewed and extended the explanation by adopting a unified symmetry-constraint formalism applicable to both PGE and APTE. The updated analysis is based on explicit consideration of two mirror-plane symmetries, M_a and \tilde{M}_b , which directly determine the allowed photocurrent components. To make the discussion transparent, we first examine separately the influence of a mirror plane on:

1. photogalvanic currents (PGE), and
2. photocurrents arising from the anisotropic photothermal effect (APTE).

Photogalvanic current:

PGE can be expressed as [7]:

$$j_\alpha = \sigma_{\alpha\beta\gamma} E_\beta(\omega) E_\gamma(-\omega), \quad (4)$$

where j_α is the DC photocurrent along direction α , $\sigma_{\alpha\beta\gamma}$ is the second-order photoconductivity tensor determined by the crystal's symmetry, and E_β are components of the driving electric field at frequency ω . Crystal symmetries impose strict constraints on $\sigma_{\alpha\beta\gamma}$.

We consider a system that is invariant under the mirror operation M_{xz} (mirror plane in the xz plane), which acts on real-space coordinates as

$$(x, y) \xrightarrow{M_{xz}} (x, -y). \quad (5)$$

Here, we restrict the notation to the in-plane coordinates x and y without loss of generality.

Under this reflection, the y component of any polar vector changes sign, while the x and z components remain unchanged. For the current density \mathbf{j} , this transformation reads

$$\begin{aligned} (M_{xz}\mathbf{j})_x(x, y) &= +j_x(x, -y), \\ (M_{xz}\mathbf{j})_y(x, y) &= -j_y(x, -y). \end{aligned} \quad (6)$$

Equivalently, this can be written as

$$j_x \xrightarrow{M_{xz}} j_x, \quad j_y \xrightarrow{M_{xz}} -j_y. \quad (7)$$

The same transformation applies to the electric field, such that

$$E_y \xrightarrow{M_{xz}} -E_y. \quad (8)$$

Because the mirror operation leaves the physical system invariant, the photocurrent must transform consistently under M_{xz} . While the left-hand side of Eq. 4 transforms as $j_y \rightarrow -j_y$, the right-hand side remains unchanged, since $(E_y)^2 \rightarrow (-E_y)^2$. This condition can only be satisfied for all spatial positions (x, y) if $\sigma_{y\beta\gamma} = 0$, which implies

$$\boxed{j_y = 0 \quad (\text{with } M_{xz}).} \quad (9)$$

Anisotropic photothermal effect (APTE):

We now apply the same mirror-symmetry analysis to the anisotropic photothermal effect. Assuming that the crystallographic a axis is aligned with the x direction, the photocurrent components can be written as [8]:

$$\begin{aligned} J_x(\mathbf{r}) &= -\sigma_a [\partial_x \Phi(\mathbf{r}) + S_a \partial_x T(\mathbf{r})], \\ J_y(\mathbf{r}) &= -\sigma_b [\partial_y \Phi(\mathbf{r}) + S_b \partial_y T(\mathbf{r})], \end{aligned} \quad (10)$$

where $\sigma_{a/b}$ and $S_{a/b}$ denote the electrical conductivities and Seebeck coefficients along the a and b axes, respectively.

Here, $\Phi(\mathbf{r})$ denotes the electrochemical potential induced by laser excitation, obtained from the continuity equation

$$\nabla \cdot \mathbf{J} = 0, \quad (11)$$

with the boundary conditions $\Phi \rightarrow 0$ far from the excitation region and $\mathbf{J} \cdot \mathbf{n} = 0$ at the sample edges (\mathbf{n} : local outward normal).

The total photocurrent is the sum of a diffusion term,

$$\mathbf{J}_d = -\sigma \nabla \Phi, \quad (12)$$

and a photothermal term,

$$\mathbf{J}_{ph} = -\sigma \mathbf{S} \nabla T, \quad (13)$$

which arises from laser-induced temperature gradients.

For an isotropic Gaussian temperature profile $T(\mathbf{r}) \propto e^{-r^2}$, a finite anisotropy ($S_a \neq S_b$) generates circulating current patterns in the bulk, with $|\mathbf{J}| \propto |S_a - S_b|$; examples simulated under the same formalism are shown in Fig. 2.

We again consider a mirror plane M_{xz} . Solving Eq. (11) using Eq. (10) gives

$$\sigma_a \partial_x^2 \Phi + \sigma_b \partial_y^2 \Phi = -\sigma_a S_a \partial_x^2 T - \sigma_b S_b \partial_y^2 T. \quad (14)$$

If the temperature profile is invariant under M_{xz} , the solution $\Phi(x, y)$ also obeys the same mirror symmetry:

$$T(x, -y) = T(x, y), \quad \Phi(x, -y) = \Phi(x, y). \quad (15)$$

[REDACTED]

Figure 2: **Simulated photocurrent distributions under different excitation conditions.** Adapted from Ref. [8], with simulations performed using the same theoretical formalism and boundary conditions. Simulated current distributions assuming a Gaussian temperature profile $T(\mathbf{r}) \propto e^{-r^2}$ following laser excitation, with $S_a \neq S_b$ and $\sigma_a = \sigma_b$. Different excitation conditions are considered, including bulk excitation (a) and excitation at edges oriented along the $\langle 100 \rangle$ (a axis) (b) and $\langle 110 \rangle$ (c) directions. The radial flow component $\mathbf{J} \cdot \hat{\mathbf{r}}$ is shown as a false-colour map. The electrochemical potential Φ is obtained numerically using Python.

Differentiation then yields

$$\begin{aligned} \partial_x \Phi(x, -y) &= \partial_x \Phi(x, y), & \partial_x T(x, -y) &= \partial_x T(x, y), \\ \partial_y \Phi(x, -y) &= -\partial_y \Phi(x, y), & \partial_y T(x, -y) &= -\partial_y T(x, y). \end{aligned} \tag{16}$$

Substituting into Eq. (10) gives the mirror-symmetry relations:

$$\boxed{\begin{aligned} J_x(x, -y) &= J_x(x, y), \\ J_y(x, -y) &= -J_y(x, y). \end{aligned}} \tag{17}$$

Equation (17) shows that the presence of the M_{xz} mirror plane imposes a clear constraint on the APTE photocurrent distribution: while it does not locally forbid a given current component, it enforces symmetry of the spatial pattern across the mirror plane. Consequently, the net photocurrent measured by integrating local currents along a detection direction will reflect these symmetry restrictions.

Summary of mirror-plane symmetry constraints and application to WTe_2 crystal

Combining Eq. (9) (PGE) and Eq. (17) (APTE), we arrive at a general rule applicable to both mechanisms: *for a system invariant under a given mirror plane, the net photocurrent—defined as the spatial integral of the local current density—can only have components lying within that mirror plane.*

In the T_d phase of WTe_2 (space group $Pmn2_1$), two in-plane mirror symmetries are relevant for photocurrent analysis:

1. The mirror plane M_a parallel to the a axis, present within each single crystal layer.
2. The glide mirror plane \tilde{M}_b parallel to the b axis, which results from the multilayer stacking arrangement.

Whether these symmetries are preserved or broken depends on the measurement geometry, and this dictates the photocurrent components allowed by symmetry. Three representative cases are summarised in Table 2:

1. In the bulk region, the presence of both mirror planes forbids any net photocurrent.

2. For edges parallel to the crystal axes, the allowed net photocurrent is along the mirror-symmetric direction, i.e., perpendicular to the edge.
3. For edges misaligned with the crystal axes, no mirror-plane constraints apply. The allowed net photocurrent can have both perpendicular and parallel components with respect to the edge.

In this way, the observed photocurrents, which are maximized under condition (3), could originate from both thermal and photogalvanic origins. For this reason, we describe these in the manuscript as “directional photocurrents”.

Table 2: Symmetry constraints on photocurrent generation in T_d -phase WTe_2 .

Excitation position	Preserved mirror symmetry	Allowed net photocurrent
Bulk	M_a and \tilde{M}_b	Forbidden
Edge \parallel crystal axis	M_a or \tilde{M}_b perpendicular to edge	Perpendicular to edge only
Generic edge	No mirror plane preserved	Parallel and perpendicular to edge

(3) The photocurrent term ought to incorporate the time term, and the rate of time decay is associated with the terahertz spectrum. Nevertheless, the model indicates that the photocurrent term excited by the light pulse remains a constant value and does not vary with time. Then, is this simplified portrayal of the actual transient photocurrent truly reasonable?

#Response#: We appreciate the referee’s careful reading, as our model is indeed in the frequency domain. The temporal dynamics of photocurrents can occur in two places - both in the decay of the directional photocurrent during propagation (eg. due to scattering effects), as well as an actual temporal decay of the driving term of the photocurrent itself. We show below that the first effect is reflected by the linewidth of the mode, and the second effect could lead to a shift in the location of the observed resonance frequency, but that this effect is small for the estimated pulse duration. Thus the first source of temporal decay is already incorporated into the model, and the second source is a minor correction to the model for the frequency range studied here.

In regard to the first effect: We acknowledge that a realistic transient photocurrent will decay in amplitude over time due to various scattering effects. Such temporal decay is analogous to damping of a harmonic oscillator. These damping effects have been included in the model (see Sec. S.8) and lead, in the Fourier domain, to a broadening of the resonance linewidth. This is indeed what we see experimentally. We have schematically illustrated the decay of the signal in time in the timing diagrams shown in Fig. S18.

A useful comparison can be made between Figs. S14 and S15. Figure S15 illustrates an idealised undamped case ($\gamma \rightarrow 0$; no amplitude decrease in time) in which the resonance peaks reduce to delta functions. When the simulation is adjusted to match experimental conditions, a finite damping ratio is introduced, corresponding to a decay in the photocurrent amplitude over time. In the frequency domain, this temporal decay manifests as a finite linewidth of the resonance (see Fig. 4b and Fig. S14), in agreement with our measured spectra. Thus, while formulated in the frequency domain, the model implicitly captures the decay of the photocurrent due to lossy propagation effects. In this way, the first source of temporal dynamics has already been incorporated into the model.

The second source of temporal dynamics is the temporal envelope of the initially-generated photocurrent. The Referee rightly notes that this effect is not present in the model; however, we show below that this effect results in a small shift of the center frequency ω_0 for the range of frequencies studied here. Ultimately, this

is the generation-side analogue of the detection-side bandwidth effect discussed above. In brief, the temporal envelope of the photocurrent generation introduces a spectral distribution to the initial photocurrent. When this spectral distribution is multiplied by the natural resonance behaviour described by the model, ω_0 is slightly shifted. Below, we assess this effect quantitatively and provide an upper boundary of <15% on the resulting shift of ω_0 (assuming scattering rates obtained from experiment).

The theoretical model proceeds analogously to determining the resonance of a damped harmonic oscillator: we solve the system for a trial frequency ω , then vary ω over a range of frequencies to obtain the full frequency response curve. By scanning ω , we determine both the resonance peak and the spectral linewidth. This is analogous to saying that the driving term is a constant in frequency, $A(\omega) = A$. Represented in the time domain, the constant-in-frequency driving term is a delta-function in time: $A(\omega) = A \leftrightarrow A(t) = \delta(t)$. Here, we neglect normalisation constants for simplicity. This represents a single impulsive kick to the system. In this way, the model is purely solving for the natural resonance behaviour of the system.

The temporal envelope of the photocurrent generation can be encoded into the model by broadening this temporal delta-function. In the frequency domain, this results in a corresponding spectral envelope that decreases with increasing ω . Because this spectral envelope decreases with increasing ω , the center frequency ω_0 is pulled toward the lower frequencies where there is greater spectral content driving the oscillator. How much ω_0 is shifted depends upon how rapidly the spectrum decays with ω , which itself depends on the temporal broadening of the generation mechanism, mechanism, as well as the FWHM of the resonance. The more the drive is temporally broadened, the more rapid the spectral decay with ω , and the more ω_0 is shifted to lower frequency.

To provide a more explicit demonstration, we consider the standard model of a damped harmonic oscillator driven by an ultrafast force,

$$\ddot{x}(t) + 2\gamma\dot{x}(t) + \omega_0^2x(t) = f(t), \quad (18)$$

where $\gamma = \frac{b}{2m}$, $\omega_0^2 = \frac{k}{m}$, $f(t) = \frac{F(t)}{m}$.

To make a simple approximation of temporal broadening, we model the excitation as an ultrafast, exponentially decaying driving force,

$$f(t) = Ae^{-t/\tau}u(t), \quad \alpha > 0, \quad (19)$$

where $u(t)$ is the Heaviside step function and τ characterises the temporal duration of the excitation.

We solve the response in the frequency domain by taking the Fourier transform, $X(\omega) = \mathcal{F}\{x(t)\}$. Using $\mathcal{F}\{\dot{x}(t)\} = i\omega X(\omega)$, $\mathcal{F}\{\ddot{x}(t)\} = -\omega^2 X(\omega)$, the Fourier transform of the driving force is

$$\int_{-\infty}^{\infty} Ae^{-t/\tau}u(t)e^{-i\omega t} dt = A \int_0^{\infty} e^{-(\tau^{-1}+i\omega)t} dt = \frac{A}{\tau^{-1} + i\omega}. \quad (20)$$

The frequency-domain response is therefore

$$X(\omega) = \frac{A}{(\tau^{-1} + i\omega)(\omega_0^2 - \omega^2 + 2i\gamma\omega)}. \quad (21)$$

In this expression, γ represents the intrinsic decay rate of the oscillator, which in our system corresponds to the decay rate of the plasmonic mode. τ represents the timescale of the current that is the driving force for the harmonic oscillator. The duration of this drive is determined by the convolution of the ultrafast laser pulse duration photoexciting the device (approximately 100 fs,) and the dynamics of directional photocurrent. From Fig. S10, we extracted a signal rise time of ~ 1.3 ps for the directional photocurrent (whose timescale

Figure 3: **Simulated normalised peak position ($\text{Im}(\chi)$) of a simple harmonic oscillator with a time-dependent, decaying driving force.** (a)-(d) plots the normalised $\text{Im}[\chi]$ for each device, assuming a time-dependent driven simple harmonic oscillator defined by Eq.21, where ω_0 is obtained from the model values plotted in Fig.4b, and γ is extracted from experimental fits. A range of τ values are simulated. An extreme upper bound on τ in these experiments is estimated as $\tau = 1.3$ ps, extracted from time-domain fits to the data as shown in Supplementary section S.5.

is also convolved there with the detection bandwidth and any dispersion that occurs between the current generated in the sample and the readout at the second switch). Thus, we can take 1.3 ps as an *extreme upper bound* of the driving force time-duration.

In Fig.3, we plot eq. 21 for a range of τ values, and for each sample (taking the undamped resonance frequency position derived from the theoretical model as shown in Fig. 4b, and the maximum scattering rate observed on each sample extracted from fits to the data). In each simulation then, $\tau = 1.3$ is an extreme *upper limit* for the effect of a time-dependent decay of the driving force. From these simulations, the peaks shift by 14%, 7%, 11%, and 13% for devices A-D respectively, when comparing the peak position with $\tau = 1.3$ ps to that with $\tau = 0.1$ ps. A major shift of ω_0 is only possible if the driving force sets a dominant time scale of $1/\tau \ll \omega_0$, as illustrated by the case of $\tau = 10$ ps in Fig. 3; however, this is certainly not the case in our measurements. Thus while the temporal dynamics of the driving force produces a small shift of ω_0 , the simple damped harmonic oscillator model provides an excellent estimate of the resonance behaviour for realistic values of τ . Finally, we note that because the frequency-dependent drive results in an apparent red-shift of the peak, these shifts of the modelled ω_0 to slightly lower frequencies actually tends to improve the agreement between model and experiment.

In our case, the good agreement for Devices A–C shows that the simplified frequency-domain model captures the dominant spectral feature — the resonance peak — even under mildly nonlinear experimental conditions, and accounting for the limited experimental bandwidth could account for the lower-frequency experimentally detected peak as compared to theory for device D. An additional decay envelope in the model would result in linewidth broadening, which is indeed what we observe experimentally. However, fully incorporating this damping envelope would add considerable complexity (such as validating a particular analytical form for the temporal envelope for all fluences) and is beyond the present scope of this work. Nevertheless, we do agree that this is an intriguing area of future work in systems where the time-domain decay is expected to play a larger role in the dynamics of the driven system.

We have now included the above discussion into the supplement, and thank the referee for bringing this point to light.

I noted that other reviewers have also raised similar questions. They noted that while the experimental results presented in this manuscript are interesting, some theoretical explanations are confusing and unconvincing. I share the reviewers' skepticism toward the speculative mechanistic explanations proposed by the authors.

#Response#: We thank the referee once again for the helpful points and suggestions. In response, we have made corrections and modifications to the manuscript. All changes are marked with text in red in the revised version. We will be happy to address any further comments or questions, and we look forward to the referee's continued feedback.

References

- [1] Kipp, G. *et al.* Cavity electrodynamics of van der waals heterostructures. *Nature Physics* 1–8 (2025).
- [2] Fertig, H. & Sarma, S. D. Collective modes in layered superconductors. *Physical review letters* **65**, 1482 (1990).

- [3] Fertig, H. & Sarma, S. D. Collective excitations and mode coupling in layered superconductors. *Physical Review B* **44**, 4480 (1991).
- [4] Jablan, M., Buljan, H. & Soljačić, M. Plasmonics in graphene at infrared frequencies. *Physical Review B—Condensed Matter and Materials Physics* **80**, 245435 (2009).
- [5] Basov, D., Fogler, M. & García de Abajo, F. Polaritons in van der waals materials. *Science* **354**, aag1992 (2016).
- [6] Moore, S. L. *et al.* Van der waals waveguide quantum electrodynamics probed by infrared nanophotoluminescence. *Nature Photonics* 1–7 (2025).
- [7] Wang, Q. *et al.* Robust edge photocurrent response on layered type ii weyl semimetal WTe₂. *Nature communications* **10**, 5736 (2019).
- [8] Wang, Y.-X. *et al.* Visualization of bulk and edge photocurrent flow in anisotropic weyl semimetals. *Nature Physics* **19**, 507–514 (2023).

Reviewer 4 & 5

The authors have made extensive efforts to answer the questions raised by the referees. The responses seem reasonable and I am happy to recommend the onward publication of the paper.

#Response#: We thank the referees for their careful reading of the revised manuscript and for their positive and constructive assessment. We are pleased that the referee considers the manuscript suitable for publication.